# Specificity in glycosylation of multiple flagellins by the modular and cell cycle regulated glycosyltransferase FlmG

**Silvia Ardissone[†], Nicolas Kint, Patrick H Viollier***

Department of Microbiology & Molecular Medicine, Faculty of Medicine / CMU, University of Geneva, Genève, Switzerland

**Abstract** How specificity is programmed into post-translational modification of proteins by glycosylation is poorly understood, especially for O-linked glycosylation systems. Here we reconstitute and dissect the substrate specificity underpinning the cytoplasmic O-glycosylation pathway that modifies all six flagellins, five structural and one regulatory paralog, in *Caulobacter crescentus*, a monopolarly flagellated alpha-proteobacterium. We characterize the biosynthetic pathway for the sialic acid-like sugar pseudaminic acid and show its requirement for flagellation, flagellin modification and efficient export. The cognate NeuB enzyme that condenses phosphoenolpyruvate with a hexose into pseudaminic acid is functionally interchangeable with other pseudaminic acid synthases. The previously unknown and cell cycle-regulated FlmG protein, a defining member of a new class of cytoplasmic O-glycosyltransferases, is required and sufficient for flagellin modification. The substrate specificity of FlmG is conferred by its N-terminal flagellin-binding domain. FlmG accumulates before the FlaF secretion chaperone, potentially timing flagellin modification, export, and assembly during the cell division cycle.

**\*For correspondence:**
patrick.viollier@unige.ch

**Present address:** [†]Center for Research on Intracellular Bacteria, Institute of Microbiology, University Hospital Center and University of Lausanne, Bugnon, Switzerland

## Introduction

Post-translational protein modification is essential for various facets in cellular biology, ranging from gene regulation to the organization of cellular structures. In all cases, biological function underlies the capacity to specifically identify and modify the correct target protein. Exquisite control mechanisms must be in place to ensure modification of the designated target, a feat that is more convoluted for proteins that are destined for the cell surface or the exterior, for example, for proteins that are first modified in the cytosol by dedicated glycosyltransferases (*Keys and Aebi, 2017*; *Nothaft and Szymanski, 2010*; *Valguarnera et al., 2016*). Unraveling the determinants underpinning the substrate selection is not only important for understanding the fundamentals and diversity in biological glycosylation systems but also has important translational implications for synthetic biology towards engineering recombinant glycoconjugates as vaccines or glycoproteins in other therapeutic applications (*Nothaft and Szymanski, 2010*; *Vimr et al., 2004*; *Cuccui and Wren, 2015*; *Ghaderi et al., 2012*).

In bacteria, extracellular proteinaceous surface structures including pili, flagella, and autotransporters as well as toxins are often post-translationally modified by glycosylation (*Nothaft and Szymanski, 2010*; *Valguarnera et al., 2016*; *Vimr et al., 2004*; *De Maayer and Cowan, 2016*; *Miller et al., 2008*; *Szymanski et al., 2003*; *Schäffer and Messner, 2017*; *Goon et al., 2003*; *Schirm et al., 2003*; *Shen et al., 2006*; *Sulzenbacher et al., 2018*; *Lu et al., 2014*). Since pili and flagella may be exposed to immune surveillance systems of eukaryotic cells, glycosylation of the structural subunits of these appendages, the pilin or flagellin, is often linked to virulence and evasion from the host immune system by molecular mimicry (*Nothaft and Szymanski, 2010*; *Schäffer and Messner, 2017*; *Arora et al., 2005*; *Logan, 2006*). In *Salmonella enterica* serovar Typhimurium

another type of flagellin modification, methylation, was recently shown to promote adhesion to host cells (*Horstmann et al., 2020*). Flagellin glycosylation may potentially affect flagellar motility in many bacterial lineages since genomic and mass spectrometry data reveal that glycosylation systems are not restricted to pathogens but also occur in non-pathogenic bacteria found in the environment (*De Maayer and Cowan, 2016*; *Schirm et al., 2005*). In several polarly flagellated Gram-negative bacteria, flagellin glycosylation is required for assembly of the flagellar filament. In *Campylobacter jejuni* and *Helicobacter pylori*, two epsilon-proteobacteria that cause a broad range of human and animal diseases, glycosylation is required for flagellar assembly, motility, and virulence (*Schirm et al., 2003*; *Linton et al., 2000*; *Guerry et al., 2006*; *Zebian et al., 2016*). *H. pylori* has a monopolar flagellum, while *C. jejuni* is bipolarly flagellated (*Kostrzynska et al., 1991*; *Guerry et al., 1991*).

In *Campylobacter* species, the exact chemical nature of glycosylation is variable but generally a nine-carbon sugar related to sialic acids such as a pseudaminic acid or legionaminic acid derivative is appended to the flagellin (*Thibault et al., 2001*; *Logan et al., 2002*). Many *Campylobacter* strains possess three dedicated NeuB-like synthases: one for sialic acid (incorporated into the lipo-oligosaccharide), one for legionaminic acid, and one for pseudaminic acid, both used to modify flagellins (*Linton et al., 2000*; *Sundaram et al., 2004*; *Chou et al., 2005*; *McNally et al., 2006*; *McNally et al., 2007*; *Schoenhofen et al., 2009*). By contrast, *Helicobacter* species seem to use pseudaminic acid only for flagellin glycosylation (*McNally et al., 2006*; *McNally et al., 2007*; *Schoenhofen et al., 2006*). In both *C. jejuni* and *H. pylori*, loss of pseudaminic acid biosynthesis results in non-motile strains lacking flagella. The abundance of intracellular flagellin is severely reduced in these mutants and the flagellins showed increased mobility by SDS-PAGE (polyacrylamide gel electrophoresis), consistent with the loss of glycosylation (*Schirm et al., 2003*; *Linton et al., 2000*). Similarly, polar flagellation in the gamma-proteobacteria such as pathogenic *Aeromonas* spp and the non-pathogenic environmental bacterium *Shewanella oneidensis* depends on glycosylation of flagellin with pseudaminic acid and another nonulosonic acid derivative, respectively (*Sun et al., 2013*; *Schirm et al., 2005*; *Wilhelms et al., 2012*). Interestingly, pseudaminic acid is also a component of surface polysaccharides such as the O-antigen of lipopolysaccharide (LPS) in *Aeromonas caviae* or the capsular polysaccharide (K antigen) in the symbiotic alpha-proteobacterium *Sinorhizobium fredii* NGR234 (*Forsberg and Reuhs, 1997*; *Le Quéré et al., 2006*; *Margaret et al., 2012*). In *A. caviae,* the genes required for pseudaminic acid biosynthesis are encoded in the O-antigen cluster and their mutation affects both flagellum and LPS O-antigen biosynthesis (*Canals et al., 2007*; *Tabei et al., 2009*).

The basis for substrate specificity in protein glycosylation systems is poorly understood and hampers biotechnological exploitation of these protein modification systems for therapeutic purposes. Flagellin glycosylation occurs at serine or threonine residues by O-linking glycosyltransferases (henceforth OGTs) that modify their substrates to various extent for each flagellin system, ranging from modification at a single site for *Burkholderia* and *Listeria* species (*Shen et al., 2006*; *Scott et al., 2011*; *Hanuszkiewicz et al., 2014*) to promiscuous modification at 19 serine or threonine residues for the *C. jejuni* flagellin (*Schirm et al., 2005*; *Thibault et al., 2001*). The modification usually occurs at the two surface-exposed central domains of flagellin, ideally positioned to influence the immunogenicity of the filament and the virulence in pathogens (*Arora et al., 2005*; *Verma et al., 2005*). Since no consensus sequence determinant in the primary structure of the flagellin acceptor (apart from the serine or threonine modification site) has been identified (*Thibault et al., 2001*), OGTs likely recognize the tertiary structure of the glycosyl acceptor in a highly specific manner. Evidence has been provided that glycosylation precedes secretion of the flagellin (*Parker et al., 2014*) via the flagellar export machinery to the tip of the growing flagellar filament (*Chevance and Hughes, 2008*). Thus, flagellin identification and subsequent glycosylation by the OGT must occur in the cytoplasm, presumably by soluble proteins. During flagellar assembly in Gram-negative (diderm) bacteria, the basal body harboring the export apparatus is assembled first in the cytoplasmic membrane, followed by envelope-spanning structures along with the external hook structure that serves as universal joint between the flagellar filament and the envelope-spanning parts (*Chevance and Hughes, 2008*). The flagellins are assembled last by polymerization on the hook into the flagellar filament (*Figure 1A*). They are usually the last proteins to be expressed and secreted during assembly, relying on temporal control mechanisms of gene expression promoting the orderly assembly of the flagellum. A key feature of polarly or bipolarly flagellated bacteria is

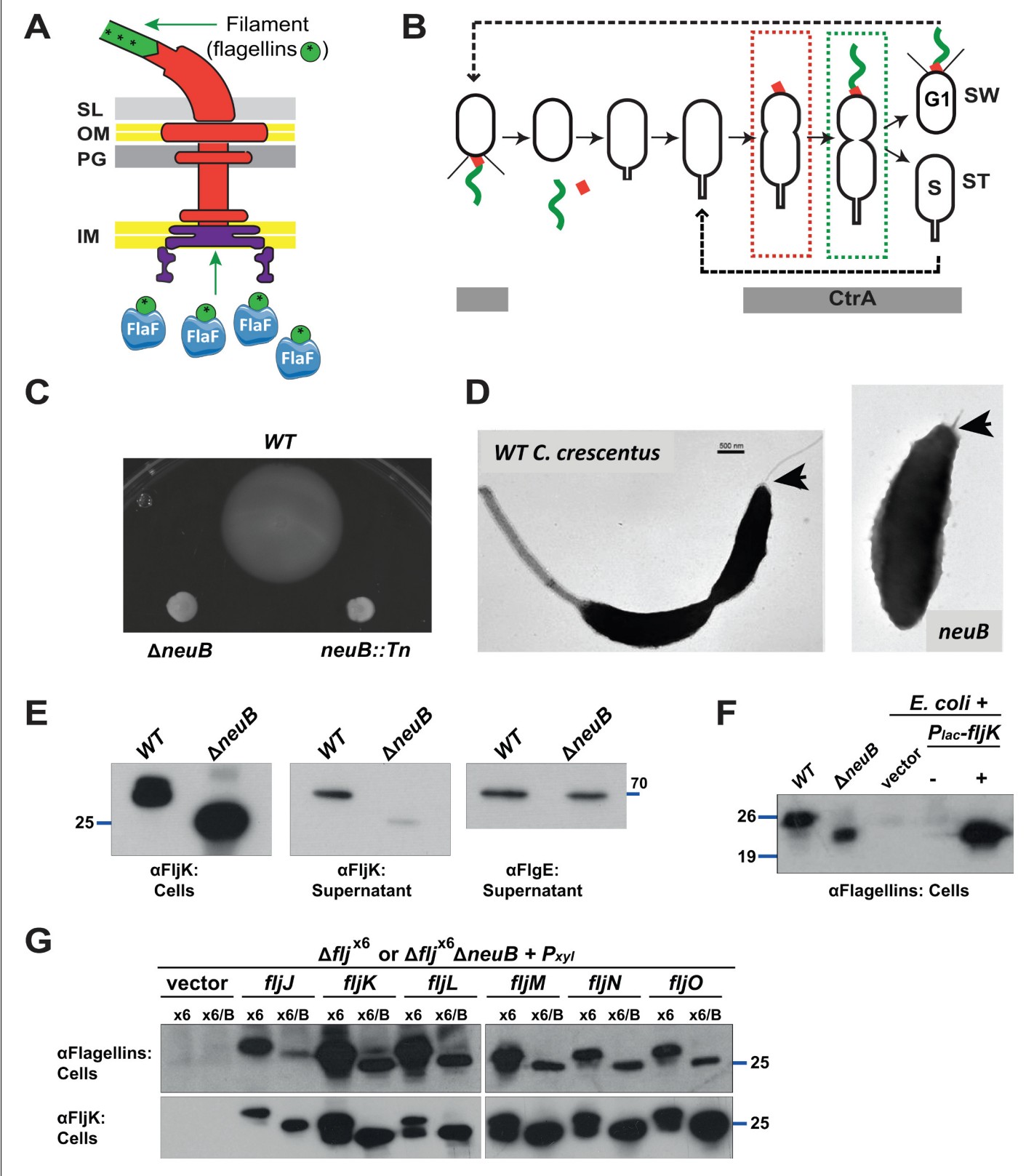

**Figure 1.** Mutation of *neuB* affects the assembly of the flagellar filament. (**A**) Schematic of the *C. crescentus* flagellum with the MS- and C-ring structures in the inner membrane (IM), the hook basal body components spanning the periplasm (with the peptidoglycan – PG – layer) and outer membrane (OM), and the filament. Flagellin subunits (in green) are brought to the export machinery by the secretion chaperone FlaF. Purple star on

*Figure 1 continued on next page*

*Figure 1 continued*

flagellins indicates the post-translational modification by glycosylation. (B) Schematic of the *C. crescentus* cell cycle. The grey bar represents the time during the cell cycle when CtrA is present and activate transcription of flagellar genes. The hook structure (FlgE, in red) is synthesized in early pre-divisional cells, whereas the flagellar filament (in green) is polymerized from flagellins in late pre-divisional cells. Both,flagellar filament and hook, are shed during the swarmer (SW) to stalked (ST) cell transition. (C) Motility assay of *neuB*::Tn and Δ*neuB* mutants compared to *WT* strain. Overnight cultures were spotted on PYE soft agar plates and incubated for 72 hours at 30°C. Compact swarms indicate that *neuB* mutant cells are non-motile. (D) *WT* and *neuB*::Tn cells analyzed by transmission electron microscopy show that only a short protrusion is visible at the SW pole of *neuB*::Tn cells, in contrast to the *WT* strain (black arrow). The images suggest that in *neuB* mutant cells the flagellar hook is stably assembled, but not the flagellar filament. (E) Immunoblots o performed with anti-FljK (αFljK, raised against FljK produced in *E. coli*, see methods) and anti-FlgE (*Hahnenberger and Shapiro, 1987*) antibodies on cell lysates and supernatants of *WT* and Δ*neuB* cultures show that flagellins are produced in Δ*neuB* cells but not efficiently exported, whereas the export of the FlgE hook protein is not affected. The migration of FljK in Δ*neuB* cells is shifted towards lower molecular mass, suggesting that post-translational modification of flagellin is defective in the Δ*neuB* mutant. Molecular size standards are indicated by the blue lines with the corresponding value in kDa. (F) Immunoblot performed on extracts from *E. coli* cells expressing *C. crescentus* FljK under control of P*lac* from a plasmid. FljK expressed in *E. coli* shows the same migration profile as in Δ*neuB* cells, indicating that *E. coli* cells cannot post-translationally modify *C. crescentus* FljK. Molecular size standards are indicated by the blue lines with the corresponding value in kDa. Note that antibodies used in this immunoblot were raised against flagellins purified from *C. crescentus* (αFlagellins; *Hahnenberger and Shapiro, 1987*). (G) Immunoblots on extracts from Δ*flj*^x6^ (x6) and Δ*flj*^x6^Δ*neuB* (x6/B) cells expressing each flagellin from a plasmid under P*xyl* control. The immunoblots were performed with antibodies raised against purified *C. crescentus* flagellins (upper panel) or FljK expressed and purified from *E. coli* (lower panel). In both cases, all six flagellins show a shift to a lower molecular mass in their migration in the absence of *neuB*, suggesting that all six flagellins are post-translationally modified. Molecular size standards are indicated by the blue lines with the corresponding value in kDa. Both antibodies recognize all six flagellins. Note that antibodies raised against flagellins purified from *Caulobacter* recognize the glycosylated form of all six flagellins better, whereas the antibodies raised against FljK expressed and purified from *E. coli* also efficiently recognizes unglycosylated flagellins.

that they must assemble a new flagellum each cell cycle. Thus, flagellar assembly, including potentially flagellin glycosylation, must be cell cycle regulated, but this remains unexplored.

The non-pathogenic and polarized alpha-proteobacterium *C. crescentus* serves as a model system to study how flagellation is regulated in space and as a function of the cell cycle (*Skerker and Laub, 2004*; *Ardissone and Viollier, 2015*). *C. crescentus* assembles a single flagellum at the new-born cell pole each cell cycle and then divides asymmetrically into a flagellated but non-replicative dispersal (swarmer, SW) cell that resides in a G1-like phase, and a capsulated sessile (stalked, ST) cell that engages in DNA synthesis (S-phase) and harbors the defining stalked appendage at the old cell pole (*Figure 1B*). Each SW cell undergoes a metamorphosis into a ST cell, replacing the flagellum with a stalk, a cylindrical extension of the cell envelope. In doing so, the flagellar filament and hook are released during the SW to ST cell transition while stalk outgrowth commences. Concurrently, replication competence is acquired and gene expression is reprogrammed toward the production of a SW daughter cell (*Laub et al., 2007*) that assembles a new polar flagellum in strict coordination with cell cycle progression.

Multiple spatiotemporal cell cycle control mechanisms feed into flagellar assembly in *C. crescentus* (*Ardissone and Viollier, 2015*). First, spatial cues that direct assembly of the flagellum to the proper site are deposited during cell division in the preceding cell cycle and inherited by the progeny (*Huitema et al., 2006*; *Lam et al., 2006*). Next, in S-phase, the transcriptional cell cycle activator CtrA and the TipF flagellar assembly organizer are expressed (*Huitema et al., 2006*; *Davis et al., 2013*; *Quon et al., 1996*; *Holtzendorff et al., 2004*; *Fioravanti et al., 2013*). CtrA induces transcription of early flagellar structural genes and regulators of late flagellar gene expression including the flagellin genes (*Quon et al., 1996*; *Stephens and Shapiro, 1993*; *Laub et al., 2002*; *Fumeaux et al., 2014*; *Fiebig et al., 2014*). Once a functional flagellar secretion structure has been assembled, the newly synthesized flagellins are exported by the FlaF secretion chaperone (*Ardissone et al., 2020*; *Llewellyn et al., 2005*; *Figure 1A*).

Here we identify, reconstitute, and dissect the substrate specificity of the O-linked flagellin glycosylation pathway of *C. crescentus*. A peculiarity of *C. crescentus* is that it expresses six flagellin paralogs (*Nierman et al., 2001*; *Faulds-Pain et al., 2011*): five structural flagellins (FljKLMNO) each of which is sufficient for flagellar filament formation and motility, while the regulatory FljJ flagellin controls translation of the others (*Ardissone et al., 2020*) but cannot support filament formation and motility in the absence of other flagellins (*Faulds-Pain et al., 2011*). We show that all six flagellins in *C. crescentus* are glycosylated in a manner that requires pseudaminic acid as donor in a reaction catalyzed by the newly identified soluble OGT FlmG. Reconstitution of FlmG-dependent flagellin

glycosylation in two heterologous systems, *S. fredii* NGR234 that naturally produces pseudaminic acid to incorporate it into the K-antigen capsule and *Escherichia coli* K12 cells engineered to produce pseudaminic acid from *C. crescentus* enzymes, reveals that FlmG is sufficient for flagellin glycosylation. The underlying specificity of glycosylation resides in the modular organization of FlmG: an N-terminal substrate (flagellin) binding domain and a C-terminal glycosyltransferase domain. We show that both domains are required for flagellin glycosylation, formation of the flagellar filament, and motility, but not for flagellin export. Finally, our studies reveal how flagellin glycosylation is tuned with progression of the *C. crescentus* cell cycle to ensure that glycosylation by FlmG can occur as soon as flagellin is translated, potentially avoiding competition for flagellin binding by the FlaF secretion chaperone (*Figure 1A*).

## Results

### NeuB is required for flagellar filament assembly

Our previously assembled library of *C. crescentus* transposon (Tn) motility mutants (*Huitema et al., 2006*) included four mutants each harboring a Tn insertion in the uncharacterized gene *CCNA_02961*, predicted to encode a NeuB-like sialic acid synthase (henceforth *neuB*). Three Tn mutants harbor a *himar1* insertion (NS7, NS44, and NS388) at different locations in *neuB*, while in the other (NS150) *neuB* is disrupted by an Ez-Tn*5* insertion. All four mutants are non-motile on soft (0.3%) agar plates and do not swim when observed by phase contrast light microscopy. An in-frame deletion of *neuB* (Δ*neuB*) recapitulated the motility defect of the Tn insertions (*Figure 1C*). The expression of NeuB from a plasmid (pMT335 [*Thanbichler et al., 2007*], see below) corrected the motility defect of Δ*neuB* cells, indicating that *neuB* function is required for motility. Transmission electron microscopy (TEM) reveale a flagellar filament on the new pole of *WT* cells, whereas Δ*neuB* cells lack a flagellar filament and only harbor a short protrusion corresponding to a hook structure (*Figure 1D*, see below). The *neuB* gene is predicted to encode a 38 kDa protein belonging to the NeuB-family of acetylneuraminate synthases (*Vimr et al., 2004*; *Linton et al., 2000*; *Chou et al., 2005*), suggesting that biosynthesis of sugars of the sialic acid family is required for flagellation *in C. crescentus*.

To gain further insights into the flagellar assembly defect of Δ*neuB* cells, we investigated whether flagellins are synthesized and exported in the absence of NeuB by immunoblotting using antibodies to the FljK flagellin (that also cross-react with other flagellins, see below). These experiments revealed lower flagellin steady-state levels in the supernatants of Δ*neuB* cells compared to *WT*. By contrast, the FlgE hook protein was present in the supernatant of both *WT* and Δ*neuB* cells to comparable levels (*Figure 1E*), in agreement with TEM analyses. Moreover, the increased migration of flagellin through SDS-PAGE suggests that the molecular mass is reduced in the absence of NeuB, consistent with NeuB-dependent post-translational modification of flagellin. In support of this conclusion, FljK expressed in *E. coli* showed the same mobility as the mobility of FljK in Δ*neuB* cells (*Figure 1F*). Next, we asked whether all six flagellins show a NeuB-dependent shift in mobility by immunoblotting and found this to be the case (*Figure 1G*). In these experiments, we expressed individual flagellins from a plasmid under the control of an inducible promoter ($P_{xyl}$) in cells deleted for all six flagellin genes (Δ*flj*$^{x6}$) and compared the mobility to that of the flagellin expressed in Δ*flj*$^{x6}$ Δ*neuB* cells. In all cases, we observed a shift to an apparent lower molecular mass in the absence of NeuB. We conclude that NeuB controls the mobility of all six flagellins.

### The pseudaminic acid synthase activity of NeuB is required for flagellation

NeuB family members are phosphoenolpyruvate (PEP)-dependent synthases that catalyze the condensation of PEP with hexoses to form sialic acid or derivatives, such as pseudaminic acid (*Vimr et al., 2004*; *Linton et al., 2000*; *Sundaram et al., 2004*; *Chou et al., 2005*; *Gunawan et al., 2005*; *Liu et al., 2009*). Some bacteria encode more than one NeuB enzyme, for example, the sialic acid synthase NeuB1 and the pseudaminic acid synthase NeuB3 from *C. jejuni* (*Linton et al., 2000*; *Chou et al., 2005*). The *Neisseria meningitidis* sialic acid synthase NeuB forms a domain-swapped homodimer, in which each monomer consists of an N-terminal TIM barrel domain similar to other PEP-utilizing enzymes and a C-terminal antifreeze-like domain (*Gunawan et al., 2005*; *Liu et al.,*

*2009*). The catalytic site is located in the C-terminal end of the TIM barrel domain, but the anti-freeze-like domain from the second monomer in the homodimer contributes key residues required for substrate binding (*Gunawan et al., 2005*). Based on primary structure alignment between NeuB from *N. meningitidis* and *C. crescentus*, we identified residues predicted to be involved in catalysis and substrate binding in *C. crescentus* NeuB (*Figure 2A*). To confirm the role of these amino acids and the requirement of NeuB catalytic activity for its function in motility and flagellin modification, we engineered single amino acid substitutions in three highly conserved residues, glutamate at position 30 (E30, implicated in the stabilization of the reaction intermediate), histidine at position 245 (H245, involved in the coordination of the $Mn^{2+}$ cofactor) and arginine at position 322 (R322, one of the residues of the antifreeze-like domain that participate in substrate binding in the active site of the second monomer in the dimer), and tested their ability to correct the motility defect of Δ*neuB* cells compared to *WT* NeuB (*Figure 2B*). The expression of NeuB variants from the vanillate-inducible $P_{van}$ promoter on pMT335 (*Thanbichler et al., 2007*) revealed that none of the variants were functional in the absence of the inducer. Under these conditions all variants, except H245A, were expressed to comparable steady-state levels as indicated by immunoblotting with polyclonal antibodies to *C. crescentus* NeuB (*Figure 2C*). In the presence of vanillate, the E30A and R322A variants still showed no activity, whereas H245A exhibited some activity in flagellin modification and motility, albeit far less than *WT* NeuB, despite accumulating to higher steady-state levels (*Figure 2B and C*). We conclude that NeuB catalytic activity is required for function in *C. crescentus.*

Next, we sought to clarify whether *C. crescentus* NeuB is a sialic acid or a pseudaminic acid synthase. To resolve this question, we conducted heterologous complementation with the three NeuB variants from *C. jejuni*, whose enzymatic activities are known: NeuB1 synthesizes sialic acid, NeuB2 produces legionaminic acid, and NeuB3 is a pseudaminic acid synthase (*Linton et al., 2000*; *Sundaram et al., 2004*; *Chou et al., 2005*; *McNally et al., 2007*; *Schoenhofen et al., 2009*). Using motility (*Figure 3A*) and flagellin modification (*Figure 3B*) as a readout for NeuB activity, we discovered that only NeuB3 can substitute for *C. crescentus* NeuB, indicating that $NeuB^{Cc}$ functions as a pseudaminic acid synthase.

Since *C. jejuni* NeuB3 also functions in the control of motility, we sought to corroborate our conclusion with a pseudaminic acid synthase that does not act in the flagellation pathway and tested whether such an enzyme can also support flagellation in *C. crescentus* Δ*neuB* cells. This experiment served to demonstrate that it is the enzymatic activity of $NeuB^{Cc}$ in pseudaminic acid synthesis that is required for flagellation in *C. crescentus*. Conversely, if *C. crescentus* NeuB is indeed a pseudaminic acid synthase, then it should be able to support pseudaminic acid synthesis in another system. We, therefore, turned to the symbiotic alpha-proteobacterium *S. fredii* NGR234 that synthetizes a K-antigen capsule composed of pseudaminic acid and glucuronic acid units (*Le Quéré et al., 2006*; *Le Quéré and Ghigo, 2009*). *S. fredii* NeuB (called RkpQ) is encoded in the K-antigen capsular polysaccharide biosynthesis locus (*rkp3*) on the pNGR234b megaplasmid (*Schmeisser et al., 2009*). First, we confirmed that *S. fredii* RkpQ was able to functionally replace NeuB in *C. crescentus*, restoring motility and flagellin migration to *C. crescentus* Δ*neuB* cells (*Figure 3A and B*), akin to NeuB3 from *C. jejuni*. To confirm that *C. crescentus* NeuB is indeed a pseudaminic acid synthase, we constructed an *rkpQ* deletion mutant (Δ*rkpQ*) in *S. fredii* and observed that this mutation blocks synthesis of the K-antigen capsule (*Figure 3C*), but not motility (*Figure 3D*). Capsule synthesis was restored by complementation of *S. fredii* Δ*rkpQ* cells with a plasmid expressing either RkpQ, *C. crescentus* NeuB or *C. jejuni* NeuB3. By contrast, *C. jejuni* NeuB1 and NeuB2 could not restore capsular polysaccharide production (*Figure 3C*). Thus, pseudaminic acid synthesis is required for motility and flagellin modification in *C. crescentus* and pseudaminic acid synthases are interchangeable.

## The OGT FlmG is required and sufficient for flagellin modification

Knowing that pseudaminic acid synthesis is required for motility and modification of all six flagellins in *C. crescentus*, we predicted that our Tn library of motility mutants should also contain Tn insertions in a gene encoding a cognate OGT. Inspection of the Tn insertion sites revealed 10 mutants with a Tn insertion in the *CCNA_01524* (henceforth *flmG*) gene: six bear a *himar1* Tn insertion at different positions in *flmG* (strains NS25, NS55, NS81, NS128, NS157, and NS192), while an Ez-Tn5 insertion disrupts *flmG* in four other mutants (NS149, NS211, NS322, and NS327). The *flmG* gene had previously been implicated in motility and flagellin biosynthesis (*Leclerc et al., 1998*; *Schoenlein et al., 1992*; *Wang et al., 1993*; *Schoenlein and Ely, 1989*; *Schoenlein et al., 1989*)

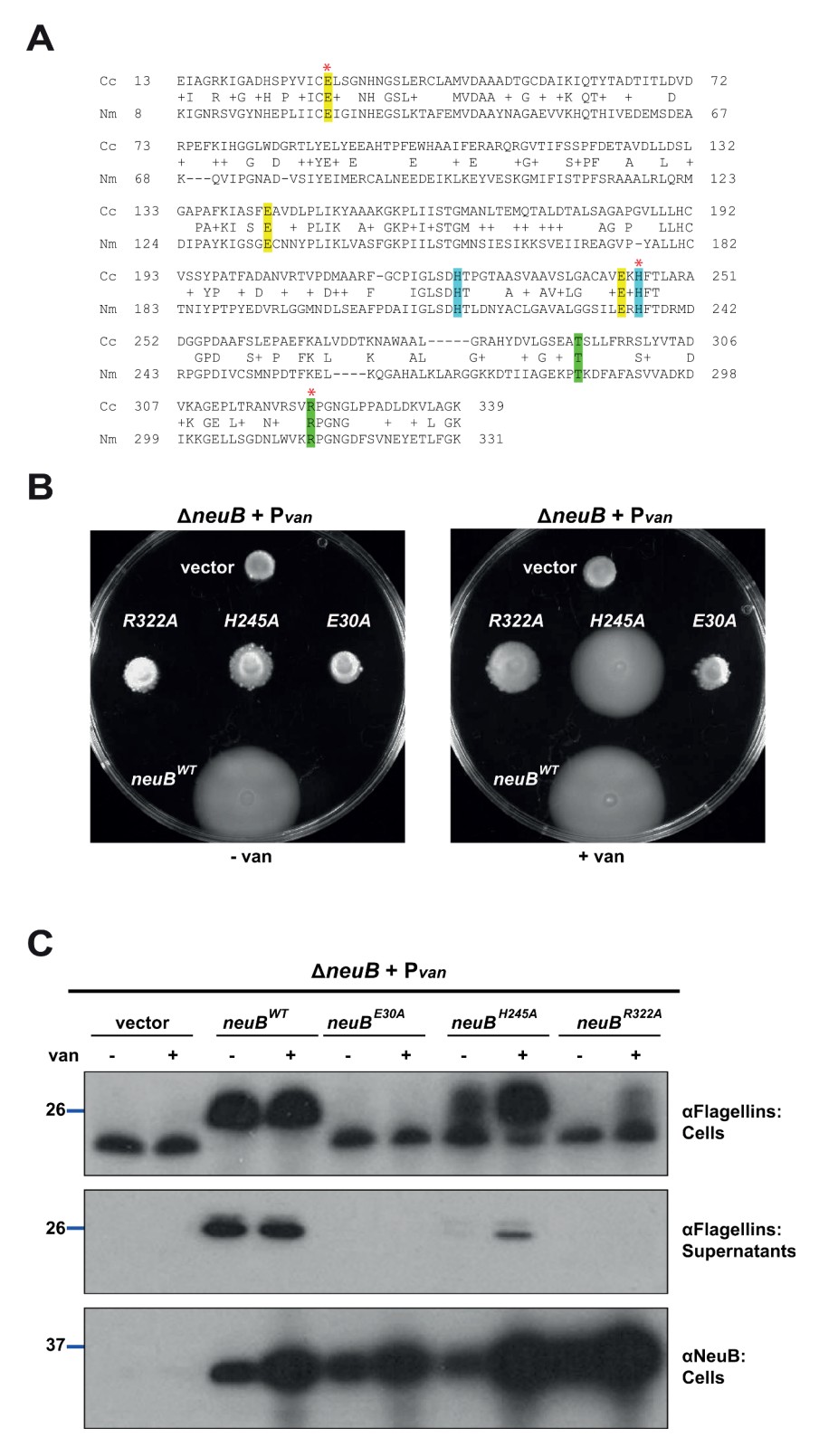

**Figure 2.** NeuB putative catalytic activity is required for motility and flagellin modification. (**A**) Sequence alignment of *C. crescentus* NeuB (Cc) to *N. meningitidis* sialic acid synthase (Nm). The three glutamate residues that have been proposed to stabilize the reaction intermediate are highlighted in yellow; the two histidine residues that coordinate the $Mn^{2+}$ cofactor are highlighted in blue; the threonine and arginine residues highlighted in green are located in the antifreeze-like C-terminal domain and protrude into the active site of the other subunit in the *N. meningitidis* sialic acid synthase

*Figure 2 continued on next page*

Figure 2 continued

dimer. The residues selected for site-directed mutagenesis in *C. crescentus* are indicated by a red asterisk. (B) Motility assay of Δ*neuB* cells complemented with different *neuB* alleles expressed from P$_{van}$ on a plasmid. Only the *WT neuB* allele can fully complement the motility defect of the Δ*neuB* strain, whereas the allele encoding NeuB(H245A) complements partially and the NeuB(E30A) and (R322A) versions do not restore motility. (C) Immunoblots showing the levels of flagellins and NeuB in Δ*neuB* cells complemented with different NeuB versions expressed from P$_{van}$ on a plasmid. All the NeuB variants were expressed (lower panel). Immunoblotting for flagellins in whole cell lysates (cells, upper panel) indicates that only the *WT* NeuB version can restore the migration profile of flagellins, whereas the E30A and R322A variants are inactive and the H245A variant shows an intermediate phenotype. The middle panel shows that only upon induction of the H245A version can the flagellins be detected in the culture supernatant, in agreement with the motility assay shown in panel B. Molecular size standards are indicated by the blue lines, with the corresponding value in kDa. Note that antibodies used in this immunoblot wereraised against flagellins purified from *C. crescentus* (αFlagellins; *Hahnenberger and Shapiro, 1987*) and they detect the glycosylated version of the flagellin better than the other flagellin antiserum (αFljK, see *Figure 1*).

and is predicted to encode a 596-residue protein of 65 kDa containing an N-terminal domain (NTD) with tetratricopeptide (TPR) repeats, known to be involved in protein-protein interactions, and a C-terminal domain (CTD) resembling glycosyltransferases (GT-B superfamily). We constructed an in-frame deletion in *flmG* (Δ*flmG*) and found the resulting mutant cells have a defect in motility (*Figure 4A*) and flagellin modification (*Figure 4B*). The motility and flagellin modification defects were corrected by the expression of FlmG *in trans* from P$_{van}$ on pMT335 (*Figure 4A and B*). Thus, FlmG acts in the same pathway as NeuB as predicted for an OGT responsible for the post-translational O-glycosylation of flagellins in *C. crescentus*.

To prove that FlmG is indeed the OGT in this modification pathway, we probed for sufficiency of flagellin modification by the expression of FlmG in a heterologous system naturally producing pseudaminic acid. We therefore chose to (co-)express FljK with or without FlmG in *S. fredii* NGR234 and probed for flagellin modification by immunoblotting using antibodies to *C. crescentus* FljK (*Figure 4C*). In the absence of FlmG, FljK showed the same mobility on SDS-PAGE as in *C. crescentus* Δ*neuB* cells. However, upon co-expression of FlmG, FljK shifted to a species with higher molecular mass and identical apparent migration on SDS-PAGE to that observed for FljK in *C. crescentus WT* cells. Importantly, this shift was dependent on the presence of pseudaminic acid, since FljK co-expressed with FlmG in *S. fredii* cells lacking pseudaminic acid (Δ*rkpQ* or Δ*rkp3_013*, see below) had the same mobility by SDS-PAGE as FljK expressed in *C. crescentus* Δ*neuB* or Δ*flmG* cells, or in *WT S. fredii* cells without FlmG (*Figure 4C*). We conclude that FlmG is required and sufficient for flagellin modification in the presence of pseudaminic acid.

A major question in glycosylation is how substrate specificity is programmed into the OGT of the system. Based on the domain organization of FlmG, we reasoned that the NTD might hold the specificity determinant toward the flagellins, perhaps by directly interacting with flagellins. By contrast, the CTD might confer OGT activity, but would not function without the NTD specificity determinant. Indeed, the expression of the CTD alone did not restore motility or flagellin modification to *C. crescentus* Δ*flmG* cells (*Figure 4A and D*). We next probed for a direct interaction of FlmG NTD with flagellins using the bacterial two-hybrid assay (BACTH, *Figure 4E*). This assay is based on the functional reconstitution of the adenylate cyclase from *Bordetella pertussis*, composed of two fragments, T25 and T18 (*Karimova et al., 1998*). When two proteins of interest fused to each fragment interact, adenylate cyclase is reconstituted and produces cyclic AMP, which in turn induces the expression of the *lacZ* gene. We tested combinations of the FlmG NTD and CTD together with the flagellins FljJ, FljK, and FljM as probes. Notably, a strong interaction was observed between each of the flagellins and the FlmG NTD (TPR), but not FlmG CTD (GT, *Figure 4E*). These BACTH results along with the domain analysis show the TPR-containing NTD is required and sufficient for a specific interaction of FlmG with multiple flagellins performing structural or regulatory functions, consistent with our finding that all flagellins are modified with pseudaminic acid by FlmG (*Figure 1G*).

## Flagellin glycosylation components are expressed before the FlaF secretion chaperone

We wished to determine if FlmG and the glycosylation pathway components are cell cycle regulated. Toward this goal, we first needed to identify the other pathway components in *C. crescentus* using a combination of genetics and bioinformatics (*Figure 5A*; *Table 1*). The first two enzymes of the pathway elucidated in *C. jejuni* are PseB (UDP-N-acetylglucosamine 5-inverting 4,6-dehydratase) and

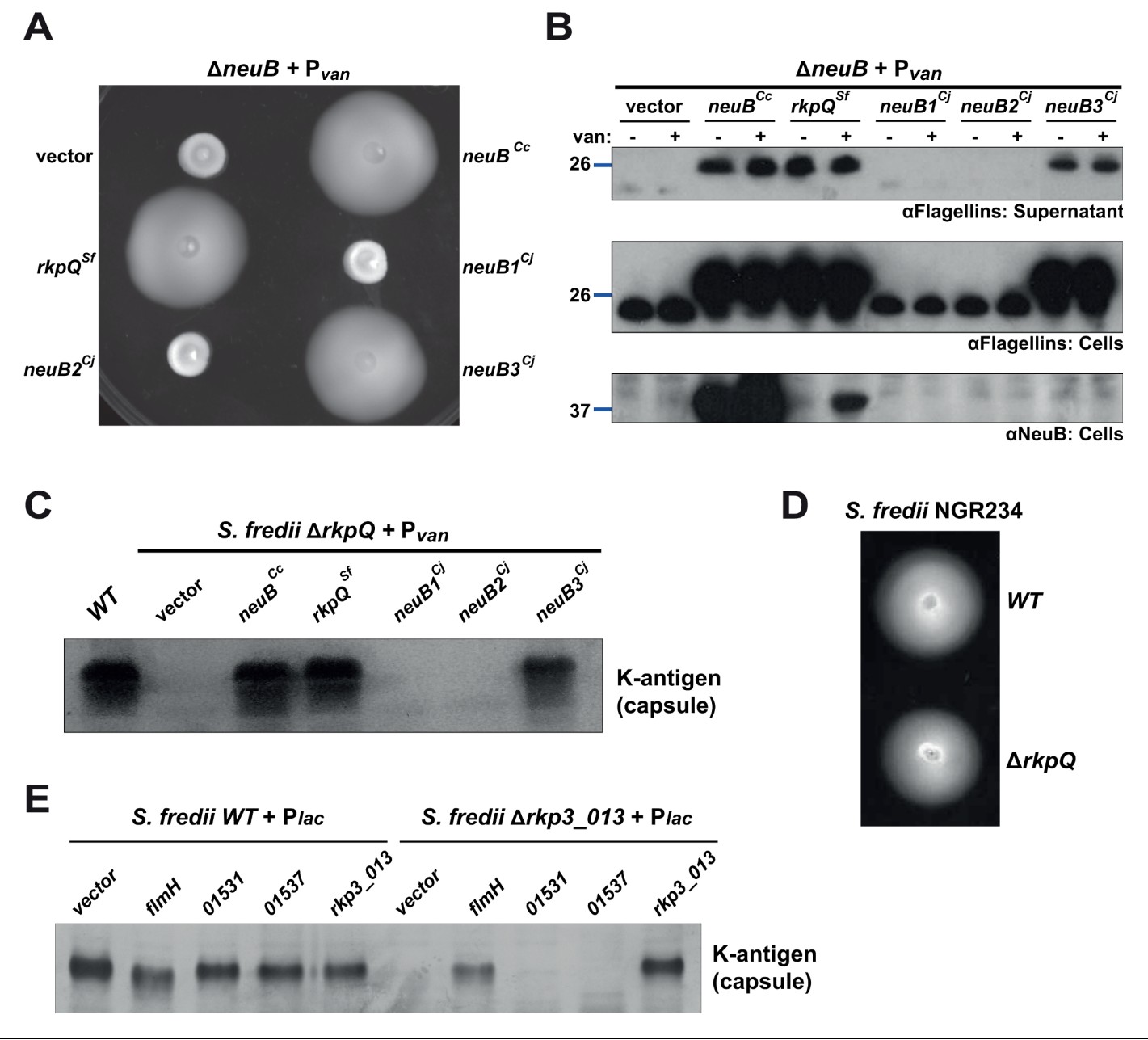

**Figure 3.** Heterologous complementation of the Δ*neuB* mutant with pseudaminic acid synthases. (**A**) Motility assay of Δ*neuB* cells complemented with different *neuB* homologs expressed from P_*van* on plasmid. Only NeuB from *C. crescentus* and the homologs known to be pseudaminic acid synthases (RkpQ from *S. fredii* and NeuB3 from *C. jejuni*) can fully complement the motility defect of the Δ*neuB* strain, whereas NeuB1 and NeuB2 from *C. jejuni* do not restore motility. (**B**) Immunoblots showing the intracellular levels of flagellins and NeuB in Δ*neuB* cells complemented with different NeuB homologs expressed from P_*van* on a plasmid. Consistent with the motility assay shown in panel A, only NeuB^Cc, RkpQ^Sf, and NeuB3^Cj can restore the flagellin migration profile (in whole cell lysates, middle panel) and the secretion of flagellin in the supernatant (upper panel) in Δ*neuB* cells. RkpQ^Sf is the protein that shows the highest similarity to NeuB^Cc, as shown by the fact that RkpQ^Sf is detected by the antibodies against NeuB^Cc (lower panel). The blue lines on the left indicate the migration of the molecular size standards, with the corresponding value in kDa. Note that antibodies used in this blot were raised against flagellins purified from *C. crescentus* (αFlagellins; *Hahnenberger and Shapiro, 1987*). (**C**) Capsular polysaccharide profile of *S. fredii* NGR234 Δ*rkpQ* mutant cells expressing different NeuB homologs from P_*van* on a plasmid. Cells with a Δ*rkpQ* mutation do not produce capsular polysaccharide, but production of capsular polysaccharide in Δ*rkpQ* cells can be restored by NeuB^Cc or NeuB3^Cj (but not by NeuB1^Cj or NeuB2^Cj), which indicates that NeuB^Cc is a pseudaminic acid synthase. (**D**) Motility assay of *S. fredii* NGR234 *WT* and Δ*rkpQ* mutant showing that mutation of the pseudaminic acid synthase does not affect motility in *S. fredii*. (**E**) Capsular polysaccharide profile of *S. fredii* WT and Δ*rkp3_013* expressing putative *C.*

*Figure 3 continued on next page*

*Figure 3 continued*

*crescentus* acetyltransferases from P*lac* on a plasmid. Production of capsular polysaccharide in Δ*rkp3_013* cells can be restored only by expression of *flmH*, which suggests that FlmH can participate in the pseudaminic acid biosynthetic pathway.

PseC (UDP-4-amino-4,6-dideoxy-N-acetyl-beta-L-altrosamine transaminase). Since genes that act in the same pathway in *C. crescentus* should be required for motility, we scanned our library of Tn mutants for insertions in orthologous genes. Indeed, the gene products of *flmA* (*CCNA_00233*) and *flmB* (*CCNA_00234*) resemble PseB and PseC, respectively. This scan revealed five mutants with Tn insertions in *flmA* (NS235, NS246, and NS294 had Hyper*Mu* insertions, NS148 harbored an Ez-Tn5 insertion and NS102 a Tn5 insertion) and three mutants with Tn insertions in *flmB* (Himar1 insertion in NS76 and Hyper*Mu* insertions in NS132 and NS255). Importantly, these mutants recapitulate the motility and flagellin modification defect of *neuB* and *flmG* mutant cells (**Figure 5B–E**) and the corresponding orthologs of *S. fredii*, RkpL and RkpM, can functionally replace *C. crescentus* FlmA and FlmB (**Figure 5B, C and E**).

For the third step of the pathway, enzymatic redundancy or promiscuity exists in *C. crescentus* as inactivation of the predicted ortholog (*flmH*, *CCNA_01523*), even with the inactivation of the paralogous genes *CCNA_01531* and *CCNA_01537* (i.e. a Δ*flmH* Δ*CCNA_01531* Δ*CCNA_01537* triple mutant), did not phenocopy the effects of *neuB*, *flmA*, *flmB,* or *flmG* single gene disruptions (**Figure 5—figure supplement 1**). Conversely, however, we demonstrated that inactivation of the *flmH* ortholog of *S. fredii* NGR234, *rkp3_013*, led to a defect in K-antigen capsule synthesis, which could be restored by the expression of *C. crescentus flmH in trans* (**Figure 3E**). Thus, FlmH can execute the corresponding acetylating step in pseudaminic acid synthesis, at least in *S. fredii*.

Bioinformatics predicts that the fourth step in pseudaminic acid biosynthesis is executed by FlmD (*CCNA_02947*) in *C. crescentus* and RkpO in *S. fredii* NGR234. To verify this prediction, we engineered an in-frame deletion in *flmD* (Δ*flmD*) and found that the resulting cells are non-motile, consistent with a previous report (**Faulds-Pain et al., 2011**), and unable to modify flagellins (**Figure 5F and G**). Importantly, we found that *S. fredii* RkpO can functionally replace FlmD, restoring motility and flagellin modification to *C. crescentus* Δ*flmD* cells (**Figure 5F and G**). Thus, the FlmD enzyme is also required for pseudaminic acid synthesis. Immediately upstream of and co-encoded with *flmD* lies *flmC* whose gene product resembles cytidylyltransferases. Since pseudaminic acid must usually be activated with cytidine 5'-monophosphate (CMP) before being incorporated into a polysaccharide or protein (**Salah Ud-Din and Roujeinikova, 2018**), FlmC likely executes this last event in *C. crescentus*. To confirm that these six enzymatic steps are necessary and sufficient for pseudaminic acid synthesis, we reconstituted FlmG-dependent glycosylation in *E. coli* K12 cells using a plasmid with a synthetic *flm* operon expressing all six enzymes (FlmA-FlmB-FlmH-FlmD-NeuB-FlmC) from open reading frames that had been codon-optimized for the expression in *E. coli*. We also introduced a second, compatible plasmid co-expressing FljK and FlmG into these cells and then probed for FljK by immunoblotting using antibodies to FljK. We indeed observed that FljK was modified under these conditions, but not in the absence of the *flm*-operon plasmid (**Figure 5H**).

Having identified the components of the flagellin glycosylation pathway, we asked whether the corresponding transcripts are cell cycle regulated. To this end we interrogated a data set of transcripts by RNA-Seq and Ribo-Seq analysis on synchronized *WT C. crescentus* cells harvested at different stages in the cell cycle. We noted that while the *flmD* and *flmC* transcripts do not fluctuate substantially in abundance during the cell cycle (**Schrader et al., 2016**), the *flmGH* and *flmAB* transcripts peak in abundance during the early pre-divisional cell stage (90 min into the cell cycle, **Figure 6A**), while the *neuB* transcript is most abundant at the late pre-divisional cell stage (after 120 min, **Figure 6A**). These findings raise the possibility that the flagellin modification genes are transcribed from a cell cycle regulated promoter. Previous chromatin-immunoprecipitation deep-sequencing (ChIP-Seq) studies revealed that the essential cell cycle regulator CtrA binds to the predicted promoter region of *flmG*, *flmA*, and *neuB* (**Fumeaux et al., 2014**; **Fiebig et al., 2014**). To demonstrate that these flagellin modification genes are indeed regulated by CtrA, we conducted promoter probe experiments with transcriptional fusions to a promoterless *lacZ* gene encoding β-galactosidase. We determined the β-galactosidase activities from these reporters in *WT* cells and *ctrA401* cells that have a missense mutation (encoding the T170I substitution) in *ctrA*, a partial loss of function mutation that renders cells temperature sensitive for growth at 37°C (**Quon et al., 1996**).

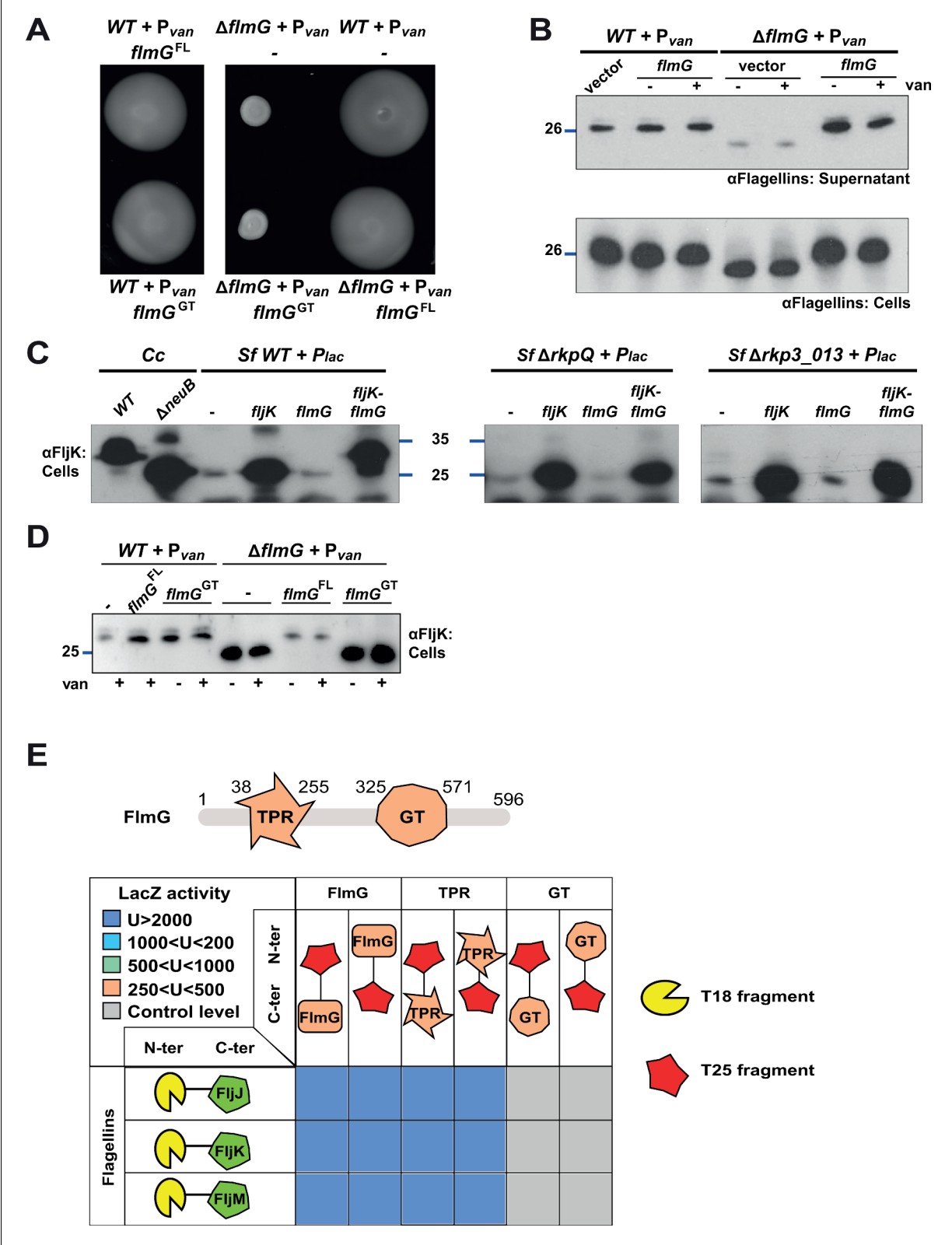

**Figure 4.** FlmG is the putative glycosyltransferase required to post-translationally modify flagellins. (**A**) Motility assay of *WT* and *ΔflmG* cells complemented with P$_{van}$-*flmG* full-length (FL) or glycosyltransferase CTD (GT) on plasmid. *ΔflmG* cells are non-motile, as indicated by the compact swarming. The expression of FlmG full-length from P$_{van}$ restores motility, in contrast to the GT domain alone. (**B**) Immunoblot showing the levels of flagellins in supernatants and whole cell lysates from *WT* and *ΔflmG* cells complemented with P$_{van}$-*flmG* on a plasmid. Flagellins produced by *ΔflmG*

*Figure 4 continued on next page*

Figure 4 continued

cells show the same migration profile as those produced by Δ*neuB* cells, with faster migration and reduced abundance in the supernatant. The blue lines on the left indicate the molecular size standard, with the corresponding value in kDa. Note that antibodies used in this blot were raised against flagellins purified from *C. crescentus* (αFlagellins; *Hahnenberger and Shapiro, 1987*). (C) Immunoblot on *C. crescentus* FljK expressed in *S. fredii* NGR234 strains. The left panel shows that when expressed alone in *WT S. fredii* FljK migrates faster, like in the *C. crescentus* Δ*neuB* mutant. By contrast, co-expression of FlmG and FljK in *S. fredii* NGR234 results in a shift in the migration profile of FljK, similar to that observed in *C. crescentus WT* cells. When FlmG and FljK are co-expressed in *S. fredii* strains unable to synthesize pseudaminic acid (Δ*rkpQ*, middle panel; Δ*rkp3_013*, right panel) FljK shows only the fast migrating band, independently from the presence of FlmG. The blue lines indicate the molecular size standards, with the corresponding value in kDa. (D) Immunoblot with anti-FljK antibodies on whole cell lysates from *WT* and Δ*flmG* cells expressing FlmG full-length (FL) or the glycosyltransferase CTD (GT) from $P_{van}$ on a plasmid. The expression of the GT domain alone does not restore the migration profile of flagellins in Δ*flmG* cells, in agreement with the motility defect shown in panel A. (E) BACTH assay showing the interaction of FlmG with flagellins. The cartoon represents FlmG, with the N-terminal TPR domain and the C-terminal glycosyltransferase domain. The scheme below shows the β-galactosidase activity, expressed in Miller units (U) of *E. coli* BTH101 cells containing the pair-wise combinations of the different constructs used for the BACTH assay. T18 and T25 correspond to the two fragments of the adenylate cyclase and are represented as yellow shape and red star, respectively. Hybrids with the proteins of interest (FlmG, FljJ, FljK, and FljM) were created as N-ter or C-ter fusions, as mentioned. In the case of FlmG, fusions were created with the full-length protein, the TPR domain or the glycosyltransferase (GT) domain only. Gray squares correspond to β-galactosidase activity similar to the control (empty plasmid, U < 250), orange squares indicate an activity between 250 and 500 U, light green squares between 500 and 1000 U, light blue between 1000 and 2000 U, and dark blue above 2000 U. The values of β-galactosidase correspond to the mean and standard deviation of three independent experiments and are listed in *Supplementary file 1* Table S1.

At the permissive temperature (30°C), *ctrA401* cells grow well, albeit several CtrA-activated promoters show substantially reduced activity (*Quon et al., 1996*; *Delaby et al., 2019*). We observed a similar reduction in activity of the promoter probe reporters of *flmG*, *flmA,* and *neuB* in *ctrA401* cells (*Figure 6B*). Conversely, a gain-of-function mutation in *ctrA* (*ctrA\**, encoding the T170A substitution) increases the activity of these reporters compared to the isogenic parent (*Figure 6B*). In these experiments, the isogenic parent is the Δ*mucR1/2* double mutant that has reduced activity of many CtrA-activated promoters. This defect can be ameliorated either by gain-of-function mutations in *ctrA* (such as *ctrA\**, *Figure 6B*) or by loss-of-function mutations in *sciP* (shown as control in *Figure 6B*), which encodes a negative regulator of a subset CtrA-activated genes (*Fumeaux et al., 2014*; *Gora et al., 2010*; *Tan et al., 2010*; *Figure 6A and B*). Thus, transcription of *flmG*, *flmA*, and *neuB* is directly integrated into the cell cycle via CtrA, as is the case for the other flagellar genes that are CtrA-dependent (directly or indirectly), including the FlaF flagellin secretion chaperone that binds all six flagellins (*Ardissone et al., 2020*) to direct their secretion and that accumulates in dividing cells (*Fumeaux et al., 2014*; *Llewellyn et al., 2005*). Immunoblotting of extracts from synchronized cells harvested at different times during the cell cycle probed with antibodies to FlmG and NeuB did not reveal substantial changes in abundance of NeuB during the *C. crescentus* cell cycle (*Figure 6—figure supplement 1A*). However, the levels of FlmG increased steadily, peaking at the end of the cell cycle (*Figure 6C*). Because FlmG is already abundant before cell division, while the FlaF secretion chaperone accumulates in dividing cells (*Llewellyn et al., 2005*), flagellin modification by FlmG can occur before the FlaF-dependent secretion and potentially avoids competition between FlmG and FlaF for flagellins.

## Discussion

### A new class of flagellin O-glycosyltransferases

There are two principal mechanisms for the transfer of sugar moieties onto an acceptor protein. An oligosaccharide synthesized on a lipid carrier can be transferred onto the acceptor protein by an oligosaccharyl-transferase (OTase)-dependent mechanism, as it occurs in the case of type IV pilin subunits in *N. meningitidis* and *Neisseria gonorrhoeae* (*Nothaft and Szymanski, 2010*; *Schäffer and Messner, 2017*; *Vik et al., 2009*; *Faridmoayer et al., 2007*). By contrast, glycosylation of flagellin subunits is OTase-independent and relies on specific glycosyltransferases that sequentially transfer monosaccharide units on the acceptor protein. Moreover, and importantly, glycosylation of flagellins occurs by a soluble glycosylation donor in the cytoplasm, the same compartment where proteins are synthesized. In the OTase-based systems the acceptor protein synthesized in the cytoplasm is (initially) spatially separated from the glycosylation donor and must be transported to the periplasmic

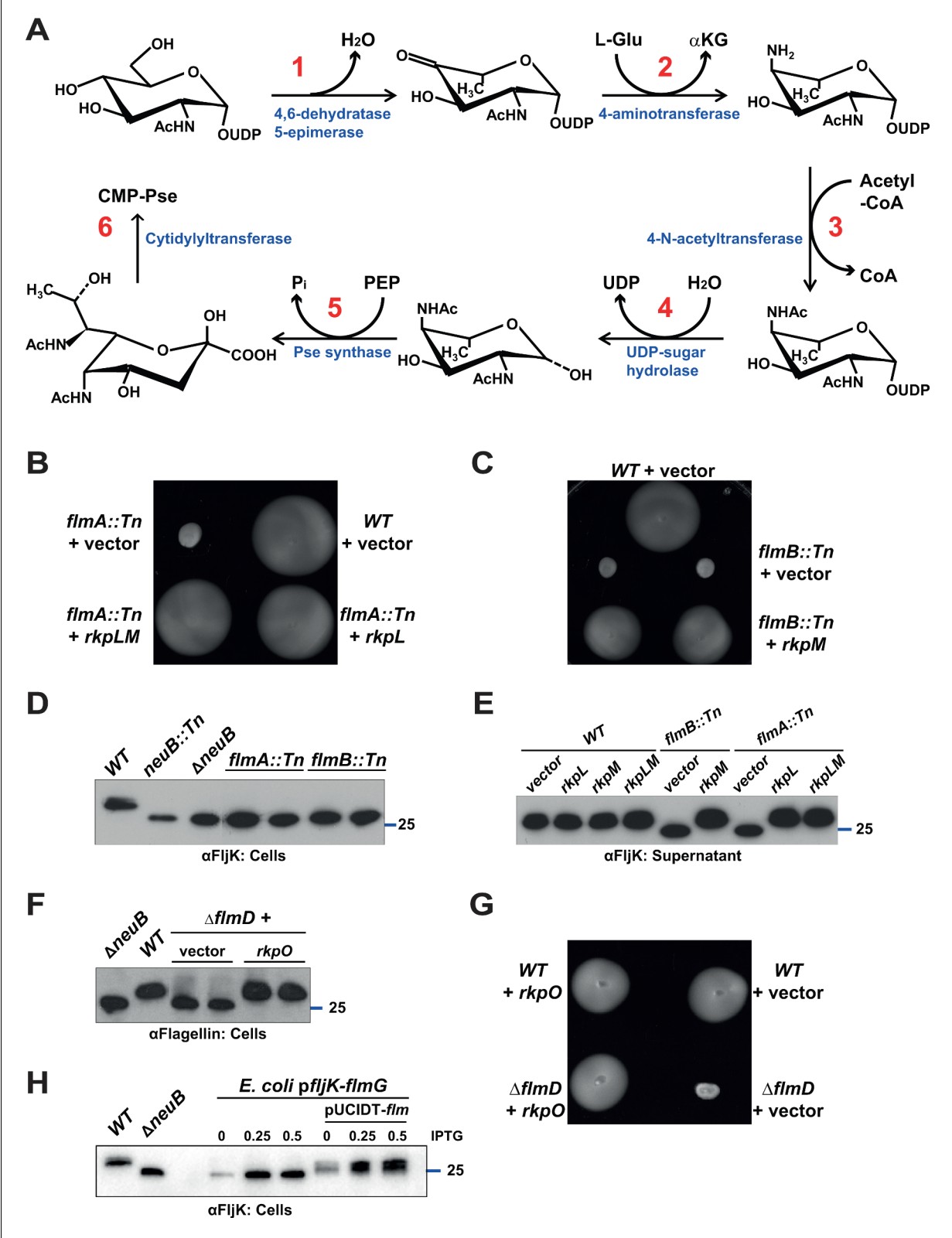

**Figure 5.** Pseudaminic acid biosynthetic pathway in *C. crescentus*. (**A**) Schematic of the pseudaminic acid biosynthetic pathway as it has been described in *C. jejuni* and *H. pylori* (reviewed in ***Salah Ud-Din and Roujeinikova, 2018***). The different steps are catalyzed by PseB (1), PseC (2), PseH (3), PseG (4), PseI (5), and PseF (6). (**B**) Motility assay of *WT* and *flmA::*Tn cells expressing *S. fredii* NGR234 *rkpL* and *rkpM* from P$_{van}$ ona plasmid. (**C**) Motility assay of *WT* and *flmB::*Tn cells expressing *S. fredii* NGR234 *rkpM* from P$_{van}$ on a plasmid. (**D**) Immunoblots of extracts from *WT* and mutant cells probed with

*Figure 5 continued on next page*

*Figure 5 continued*

polyclonal anti-FljK antibodies on whole cell lysates. Flagellins produced by *flmA::Tn* and *flmB::Tn* cells show the same migration profile as in *ΔneuB* mutant cells. The blue line indicates the migration of the molecular size standard, with the corresponding size in kDa. (E) Immunoblots of supernatants from *WT*, *flmA::Tn* and *flmB::Tn* cells probed with polyclonal anti-FljK antibodies. The expression of the *S. fredii* homologs RkpL and RkpM from P$_{van}$ on a plasmid restores the migration profile and secretion of flagellin in *flmA::Tn* and *flmB::Tn* cells, respectively. The blue line indicates the migration of the molecular size standard, with the corresponding size in kDa. (F) Immunoblot of extracts from *WT* and *ΔflmD* cells. Mutation of *flmD* impairs post-translational flagellin modification, and the defect is complemented by the expression of the *S. fredii* NGR234 homologue RkpO from a plasmid. The blue line indicates the migration of the molecular size standard, with the corresponding size in kDa. Note that antibodies used in this blot were raised against flagellins purified from *C. crescentus* (αFlagellins; *Hahnenberger and Shapiro, 1987*). (G) Motility assay of *WT* and *ΔflmD* cells expressing *S. fredii* NGR234 RkpO from P$_{van}$ on a plasmid. (H) Immunoblot with anti-FljK antibodies on whole cell lysates from *E. coli* expressing *fljK* and *flmG* from P$_{lac}$ on a plasmid, in the presence or absence of a compatible plasmid carrying the complete set of *Caulobacter* genes for the pseudaminic acid biosynthetic pathway (pUCIDT-*flm*, see Materials and methods and *Supplementary file 1* Table S3). In the absence of pUCIDT-*flm*, FljK shows the same migration profile as in *Caulobacter ΔneuB* cells, whereas in the presence of pUCIDT-*flm* FljK migration is shifted toward higher molecular mass, as in *Caulobacter WT* cells. The values above the panel indicate the concentration of the inducer for P$_{lac}$-*fljK-flmG* (mM IPTG). The blue line indicates the migration of the molecular size standard, with the corresponding size in kDa.

The online version of this article includes the following figure supplement(s) for figure 5:

**Figure supplement 1.** Acetyltransferase function is likely redundant in *Caulobacter*.

compartment for modification by the OTase, while the lipid-anchored donor must be presented on the periplasmic (extra-cytoplasmic) face of the membrane. Therefore, owing to these advantages, the flagellin glycosylation systems have great engineering potential for custom-designed O-glycosylation of proteins of interest.

The molecular basis underlying the specificity of bacterial protein glycosylation systems, especially for the cytoplasmic O-glycosylation systems, is poorly understood. Recently evidence was provided that the Maf glycosyltransferase from *Geobacillus kaustophilus* and *Clostridium botulinum* can modify flagellin with N-acetylneuraminic acid or 3-deoxy-D-manno-octulosonic acid by O-linkage at threonine and serine residues when expressed in *E. coli* (*Khairnar et al., 2020*). The Maf orthologs from *Aeromonas caviae* and *Magnetospirillum magneticum magneticum* are required for flagellin glycosylation *in vivo* (*Sulzenbacher et al., 2018*; *Parker et al., 2014*). While *C. crescentus* does not encode a Maf ortholog in its genome, we discovered that it uses FlmG, the defining member of a hitherto unknown class of OGTs, to glycosylate flagellins in hosts that can synthesize pseudaminic

**Table 1.** Homology of the *C. crescentus* enzymes involved in the pseudaminic acid biosynthesis pathway to *S. fredii* NGR234 and *C. jejuni* proteins.

| *C. crescentus* | *S. fredii* NGR234 | *C. jejuni* NCTC 11168 | Enzymatic activity |
|---|---|---|---|
| FlmA (CCNA_00233) | RkpL 52% id; 68% sim | PseB 57% id; 72% sim | Dehydratase |
| FlmB (CCNA_00234) | RkpM 44% id; 58% sim | PseC 34% id; 52% sim | Aminotransferase |
| FlmH (CCNA_01523) | Rkp3_013[†] 26% id; 37% sim | PseH[‡] 26% id; 50% sim | N-acetyltransferase |
| FlmD (CCNA02947) | RkpO 36% id; 49% sim | PseG 22% id; 41% sim | UDP-sugar hydrolase |
| NeuB (CCNA_02961) | RkpQ 56% id; 70% sim | PseI 43% id; 59% sim | Pseudaminic acid synthase |
| FlmC[§] (CCNA_02946) | RkpN | PseF | Cytidyltransferase |
| FlmG (CCNA_01524) | - | - | Protein Glycosyltransferase |

* Although *C. crescentus* genomes encodes for several putative acetyltransferases, only CCNA_01531 has significant homology to FlmH (35% identity [id] and 54% similarity [sim]).

† FlmH aligns to Rkp3_013 only for about 50% of its length.

‡ FlmH aligns to PseH for about 75% of its length.

§ FlmC shows no homology to RkpN or PseF; RkpN and PseF share 46% identity and 68% similarity.

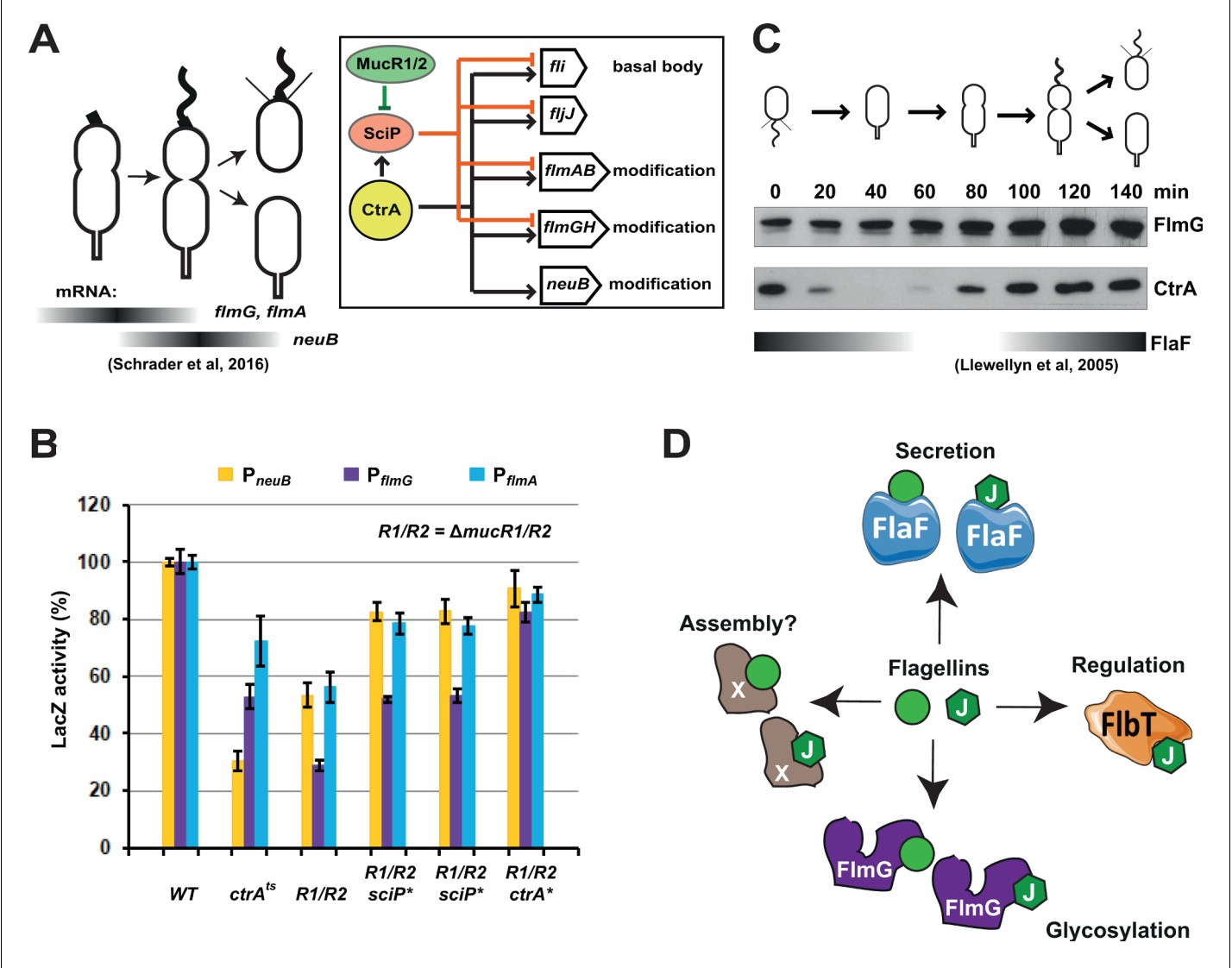

**Figure 6.** Transcription of genes encoding flagellin-modification proteins is cell cycle regulated. (**A**) Schematic of the transcriptional regulatory network controlling the expression of flagellar and pseudaminic acid biosynthetic genes in *C. crescentus* pre-divisional cells. The gray bars represent the time during the cell cycle when *flmA*, *flmG*, and *neuB* mRNA levels peak (*Schrader et al., 2016*). The transcriptional regulatory network is represented in the box: CtrA promotes the expression of genes encoding the basal body (*fli* genes), the regulatory flagellin FljJ and several components of the flagellin glycosylation pathway (*flmAB* and *flmGH* operons, *neuB*). The expression of these same genes is repressed by SciP (with the exception of *neuB*), whose expression is under control of CtrA (positively) and MucR1/2 (negatively). (**B**) β-galactosidase activity of P*neuB*, P*flmG*, and P*flmA* transcriptional fusions. Transcription of *neuB*, *flmG* and *flmAB* is significantly reduced in *ctrA401* (T170I, temperature sensitive allele *ctrA^ts^*) and double Δ*mucR1*Δ*mucR2* (*R1/R2*) strains. β-galactosidase activity of P*neuB*, P*flmG*, and P*flmA* is partially restored in Δ*mucR1*Δ*mucR2* cells carrying *ctrA(T170A)*, *sciP(T24I)*, or *sciP(T65A)* alleles (*ctrA** and *sciP**). Values are expressed as percentages (activity in *WT* NA1000 set at 100%). (**C**) Immunoblot showing FlmG and CtrA protein levels in synchronized *C. crescentus* cells. The gray bar represents the abundance of the flagellin chaperone FlaF along the cell cycle, as previously shown (*Llewellyn et al., 2005*). (**D**) Model depicting a flagellin-centric view of the events during flagellar assembly in *C. crescentus*. The FljJ flagellin is illustrated as a green hexagon, whereas the other flagellins (FljKLMNO) are depicted as green circles. The events acting on flagellins are translational regulation (binding of FljJ by FlbT), glycosylation by FlmG, secretion by FlaF and possibly assembly by an unknown factor (X).

The online version of this article includes the following figure supplement(s) for figure 6:

**Figure supplement 1.** NeuB is stable along the cell cycle and flagellins can be glycosylated in HBB mutants or in the absence of the FlaF chaperone.

acid. The very recent evidence that *C. crescentus* modifies the FljK flagellin at four threonine residues with a molecule predicted to be an O-linked glycan (*Montemayor et al., 2020*) aligns well with our finding that pseudaminic acid is required for FlmG to modify FljK. The ability of FlmG to function in a heterologous system when offered pseudaminic acid raises the question whether FlmG can also accept another (activated) nonulosonic acid, such as sialic acid, as donor for the modification reaction, or other sugars that resemble the sialic acids of humans. Our demonstration that FlmG glycosylates flagellin in *E. coli* expressing a synthetic pseudaminic acid biosynthesis (*flm*) operon from *C. crescentus* simplifies future analyses of the donor promiscuity for FlmG-dependent flagellin glycosylation, as it can now be simply achieved by the expression of orthologous sialic acid biosynthesis pathways in *E. coli*. The NeuB family of nonulosonic acid synthases enzymes may constitute a convenient marker for the identification of genomes encoding biosynthesis pathways for sialic acid-like molecules that could then subsequently be used to probe for glycosylation of flagellin or other (potentially promiscuous) acceptor proteins.

## Promiscuous flagellin binding by FlaF and FlmG

Since all six *C. crescentus* flagellins are glycosylated by FlmG, a certain degree of substrate promiscuity is inherent in the system. While the FljJ flagellin only bears 49% amino acid identity compared to the other flagellins, there are still at least 20 serine and threonine residues that are conserved between FljJ and FljK and that FlmG could potentially modify. However, only one of the four threonines that are modified in FljK is present in all six flagellins (*Montemayor et al., 2020*). We showed that the FlmG NTD binds the flagellins via the TPR region and that this domain is required for motility and efficient glycosylation (*Figure 4A, D and E*). A free-floating glycosyltransferase (CTD) in the cytoplasm cannot bind the flagellins alone in BACTH assays (*Figure 4E*) and can not modify sufficient flagellin to support assembly of the flagellar filament in *C. crescentus*. At this point, we cannot rule out that the NTD also plays a role in controlling the activity of the glycosyltransferase domain in the CTD.

Recent BACTH assays revealed that the FlaF secretion chaperone binds the six flagellins as well (*Ardissone et al., 2020*). It is expected that FlaF can bind all flagellins, since it must direct their secretion via the flagellar export apparatus to assemble into the flagellar filament. While all flagellins are assembled into the flagellar filament (*Driks et al., 1989*), FljJ is the only flagellin that cannot assemble into a flagellar filament on its own (*Faulds-Pain et al., 2011*). FljJ is the first flagellin to be synthesized and exported during the cell cycle as its location within the filament is the most proximal to the hook (*Driks et al., 1989*). As an exported flagellin, FljJ is also glycosylated like the other flagellins and it therefore must also be bound by FlmG and FlaF (*Figure 1A*). Therefore, FlmG and FlaF could potentially compete for the same substrate. However, FlmG is already present well before FlaF accumulates (*Llewellyn et al., 2005*; *Figure 6C*). Moreover, flagellin modification does occur in mutants lacking a flagellar hook basal body complex (*Figure 6—figure supplement 1B, C*) or the flagellin chaperone FlaF (*Figure 6—figure supplement 1A*), confirming that modification can occur without the secretion apparatus and the secretion chaperone.

It would be interesting to elucidate whether FlaF and FlmG bind the flagellins in the same way, and compare it to the highly specific interaction of the FljJ flagellin with *C. crescentus* FlbT (*Figure 6D*), a post-translational regulator of flagellin transcripts that only interacts with FljJ, but not other flagellins as determined by BACTH assay (*Ardissone et al., 2020*). The preference of FlbT for FljJ underlies the translational repression mechanism used to regulate the expression of transcripts encoding the five (other) structural flagellins. Translational repression by FlbT•FljJ is relieved with the removal of FljJ from the cytoplasm by secretion through the flagellar secretion conduit (hook basal body, HBB; *Chevance and Hughes, 2008*) once it has been assembled and the FlaF secretion chaperone has accumulated. Since the translational regulation by FlbT requires the presence of FljJ, translational repression is inactivated with the secretion of FljJ.

## Roles of pseudaminic acid in cell surface structures

The presence of sialic acid-like sugars on the surface of pathogenic bacteria may serve to evade the host immune system. However, flagellin glycosylation also occurs in environmental (non-pathogenic) bacteria, where the post-translational modification likely serves another purpose. In the case of *Campylobacter*, *Helicobacter*, and *Aeromonas*, the loss of flagellin glycosylation prevents assembly of a

functional flagellar filament. By contrast, in *Pseudomonas* and *Burkholderia* species, glycosylation of flagellins is not required for filament assembly (*Schirm et al., 2003*; *Linton et al., 2000*; *Scott et al., 2011*; *Parker et al., 2014*; *Taguchi et al., 2008*). In *C. crescentus*, no flagellar filament is assembled in the absence of glycosylation, although flagellins accumulate in the supernatant of Δ*neuB* or Δ*flmG* cultures. However, the level of the flagellins in the supernatant is reduced in these mutants compared to the *WT*, suggesting that non-glycosylated flagellin is less efficiently exported and/or less stable in the supernatant than modified flagellin akin to *A. caviae* (*Parker et al., 2014*).

Why pseudaminic acid is essential for flagellation in many polarly flagellated bacteria remains unclear. While the abundance of secreted flagellins is substantially reduced in the absence of glycosylation, secretion is still functional. However, we cannot rule out that the efficiency of secretion is substantially reduced and that the residual level of secreted flagellin does not suffice for assembly of a filament to support motility. However, since only hook structures are observed in glycosylation mutants rather than short flagellar filaments, it appears that filament assembly is defective. Thus, a problem other than flagellin secretion appears to underlie the flagellar assembly defect in glycosylation mutants. It is possible than an assembly factor exists that preferentially binds glycosylated flagellins upon translocation, facilitating their incorporation into the flagellar filament (*Figure 6D*). Such a factor would likely be encoded in a flagellar assembly gene cluster in polarly flagellated bacteria that glycosylate flagellin, but absent from peritrichous flagellation systems. The loss of such a factor should prevent flagellar filament formation, but neither flagellin secretion, nor glycosylation.

It is possible that glycosylation simply promotes self-assembly of the flagellins into the filament in *C. crescentus* without the aid of another factor. However, many flagellated bacteria assemble flagellar filaments without the need for glycosylation, indicating that flagellins *per se* do not depend on glycosylation for self-assembly. It is not evident why specifically the flagellins of polar flagellation systems, such as the *C. crescentus* flagellins, should have evolved a dependency on glycosylation to (self-) assemble into a filament. Indeed, the glycosylated residues do not appear to reside in the flagellin inter-subunit contact area within the flagellar filament (*Montemayor et al., 2020*). A more appealing scenario is that glycosylation serves another purpose as it does endow flagellar assembly with an additional regulatory event. Perhaps it permits differentiating flagellins from other secretion substrates present in the cytoplasm (i.e. hook or capping proteins [*Chevance and Hughes, 2008*]) and/or it permits tuning flagellin assembly with another input such as a cell cycle cue. In monopolar and bipolar flagellation systems, flagellar assembly must be highly coordinated with the cell cycle, since flagellar assembly must occur once per division cycle. Such delicate temporal control is not needed for peritrichous flagellation systems because assembly occurs in an unsynchronized fashion with each machinery being in a different assembly state as the cell elongates. Since pseudaminic acid synthesis by NeuB requires phosphoenolpyruvate (PEP), flagellation in *C. crescentus* is indirectly PEP-dependent. PEP is also well known for its regulatory functions as a phosphoryl-donor in phosphotransferase signaling (PTS) systems (*Tchieu et al., 2001*; *Pflüger-Grau and Görke, 2010*; *Postma et al., 1993*). Interestingly, a PTS system is required for the production of the alarmone (p) ppGpp (*Ronneau et al., 2016*; *Sanselicio and Viollier, 2015*; *Hallez et al., 2017*) that promotes retention of *C. crescentus* in the flagellated state. Given this linkage, it is appealing to propose that PEP-dependent glycosylation with pseudaminic acid serves to integrate flagellar assembly with (p)ppGpp-dependent cell cycle control. Interestingly, pseudaminic acid synthesis also branches from a precursor of cell wall (peptidoglycan, PG) and lipopolysaccharide (O-antigen) synthesis, UDP-N-acetyl-glucosamine, indicating that flagellin glycosylation could be linked to cell envelope modifications, for example, to ensure that the formation of the PG-based division septum is linked to the synthesis of the flagellar filament. In other environmental bacteria, such as *S. fredii* NGR234, pseudaminic acid is not required for motility (*Figure 3D*) but it is a component of the capsular K-antigen (*Le Quéré et al., 2006*), where it may still be coordinated with the cell cycle, for example, again with septation. In *C. crescentus* the capsule is cell cycle regulated (*Ardissone et al., 2014*), but it is composed of different monosaccharides (*Ardissone et al., 2014*). Pseudaminic acid or other sialic acid-like sugars synthesized by NeuB paralogs may be generally exploited for metabolic coordination of cell (cycle) processes in bacteria.

## Materials and methods

### Strains and growth conditions

*C. crescentus* NA1000 (*Evinger and Agabian, 1977*) and derivatives were grown at 30°C in PYE (peptone-yeast extract) or M2G (minimal glucose) (*Ely, 1991*). *S. fredii* NGR234 (*Stanley et al., 1988*) was grown at 30°C in TY (tryptone-yeast extract). *E. coli* S17-1 λpir (*Simon et al., 1983*), EC100D and Rosetta(DE3)pLysS (Novagen) were grown at 37°C in LB. Motility assays, swarmer cells isolation, electroporations, biparental matings, and bacteriophage φCr30-mediated generalized transductions were performed as described (*Ely, 1991*; *Chen et al., 2005*; *Viollier and Shapiro, 2003*; *Viollier and Shapiro, 2004*). Antibiotics were used at the following concentrations: nalidixic acid 20 µg/mL, kanamycin 20 µg/mL in solid medium and 5 µg/mL in liquid (50 µg/mL for *S. fredii*), gentamicin 1 µg/mL (20 µg/mL for *E. coli* and *S. fredii*), tetracycline 1 µg/mL (10 µg/mL for *E. coli* and *S. fredii*). Plasmids were introduced into *S. fredii* by bi-parental mating and into *C. crescentus* by electroporation.

### Production of antibodies and immunoblots

For the production of antibodies, His$_6$-NeuB and His$_6$-SUMO-FlmG$_{(301-500)}$ were expressed in *E. coli* Rosetta(DE3)pLysS cells and the recombinant proteins were purified using Ni-NTA agarose (Qiagen). Purified His$_6$-NeuB and His$_6$-SUMO-FlmG$_{(301-500)}$ were excised from 12.5% SDS polyacrylamide gels and used to immunize rabbits (Josman LLC, Napa, CA). For immunoblots, protein samples were separated on SDS polyacrylamide gel, transferred to polyvinylidene difluoride (PVDF) Immobilon-P membranes (Merck Millipore) and blocked in TBS (Tris-buffered saline) 0.1% Tween 20% and 5% dry milk. The anti-sera were used at the following dilutions: anti-CtrA (1:10,000; *Domian et al., 1997*), anti-FlgE (1:50,000; *Hahnenberger and Shapiro, 1987*), anti-flagellins (raised against purified flagellins from *C. crescentus*; indicated as αFlagellins in figures; 1:20,000; *Hahnenberger and Shapiro, 1987*), anti-NeuB (1:10,000), anti-FlmG (1:10,000), anti-FljK (raised against His$_6$-FljK expressed in *E. coli*; indicated as αFljK in figures; 1:10,000; *Ardissone et al., 2020*). Protein-primary antibody complexes were visualized using horseradish peroxidase-labelled anti-rabbit antibodies and ECL detection reagents (Merck Millipore).

### Site-directed mutagenesis of *neuB*

To create *neuB* alleles mutated in the predicted catalytic residues, the *neuB* ORF was sub-cloned into pOK12 (*Vieira and Messing, 1991*) as *Nde*I/*Eco*RI fragment from pMT335-*neuB*. pOK-*neuB* was used as a template for oligonucleotide site-directed mutagenesis. Two complementary oligonucleotide primers containing the desired mutation were designed for each point mutation (*Supplementary file 3* Table S3). PCR reactions were composed of 30 cycles, carried out under the following conditions: denaturation, 94°C for 1 min; annealing, 60°C for 1 min; extension, 68°C for 8 min. The PCR products were treated with *Dpn*I to digest the template DNA and used to transform *E. coli* EC100D competent cells. The constructions obtained were verified by sequencing and sub-cloned as *Nde*I/*Eco*RI fragments into pMT335.

### Transmission electron microscopy

Transmission electron microscopy (*Skerker and Shapiro, 2000*) was performed on a JEOL 1200EX with samples that were fixed in 2.5% gluteraldehyde/25 mM cacodylate-HCl buffer (pH 7.4) and negatively stained with 1% uranyl acetate for 15 min on a 300 mesh nickel formvar-coated grid stabilized with an evaporated carbon film (*Huitema et al., 2006*; *Radhakrishnan et al., 2008*).

### Capsular polysaccharide analysis by SDS-PAGE and silver staining

Strains were grown in TY supplemented with vanillate 0.5 mM or IPTG 1 mM for 24 hr. Polysaccharide extractions were made from cells collected by centrifuging 4 mL of culture. The cell pellets were resuspended in 30 µL of lysis buffer (1 M Tris-HCl [pH 6.8], 2% [wt/vol] SDS, 4% [vol/vol] β-mercaptoethanol, 10% [vol/vol] glycerol, and 0.03% [wt/vol] bromophenol blue) and boiled for 10 min. Lysed cells were treated with 10 µL of proteinase K (2.5 mg/mL) at 60°C for 3 hr, then the samples were diluted by adding 80 µL of sample buffer (120 mM Tris-HCl [pH 6.8], 3% [wt/vol] SDS, 9% [vol/vol] β-mercaptoethanol, 30% [vol/vol] glycerol, and 0.03% [wt/vol] bromophenol blue). Polysaccharides

were separated by SDS-PAGE (18% acrylamide) and stained for KPS as described (*Le Quéré et al., 2006*).

## Bacterial adenylate cyclase two-hybrid (BACTH) assays

Protein-protein interactions were studied through the BACTH system (Euromedex, France). Plasmid constructions and *E. coli* co-transformation were performed according to the manufacturer's instructions. This system is based on the reconstitution of the adenylate cyclase (Cya) from *Bordetella pertussis* which is composed of two different fragments T25 and T18. Heterodimerization of the hybrid-proteins leads to the functional complementation of the adenylate cyclase that results in the production of cAMP leading to *lacZ* gene expression. The entire open reading frames of *fljJ*, *fljK*, *fljM*, and *flmG*, as well as the N-terminal (TPR, amino acids 1–313) and C-terminal (GT, amino acids 309–596) parts of FlmG were cloned into plasmids allowing fusion with T18 or T25 fragments either in N-terminal (pKNT25/pUT18) or C-terminal position (pKT25/pUT18C). Different combinations of T18 and T25 fusion plasmids were co-transformed in the *E. coli* BTH101 Δ*cya* strain. The β-galactosidase activity was measured after an overnight culture of three independent co-transformants in LB medium as recommended by the manufacturer's instruction. Co-transformation of BTH101 strain with the empty plasmid pK(N)T25 and pUT18(C) served as a control for basal β-galactosidase activity level (around 100 Miller unit).

## β-galactosidase activity assays

β-galactosidase assays were performed at 30°C. Cells (50–200 µL) at $OD_{660nm}$ = 0.1–0.5 were lysed with 30 µL of chloroform and mixed with Z-buffer (60 mM $Na_2HPO_4$, 40 mM $NaH_2PO_4$, 10 mM KCl, and 1 mM $MgSO_4$; pH 7) to a final volume of 800 µL. The reaction was started and timed following the addition of 200 µL of ONPG (o-nitrophenyl-β-D-galactopyranoside, 4 mg/mL in 0.1 M potassium phosphate, pH 7). Upon medium-yellow color development, 400 µL of 1 M $Na_2CO_3$was added to stop the reaction. $OD_{420nm}$ of the supernatant was recorded and Miller units were calculated as follows: $U=(OD_{420nm}*1000)/(OD_{660nm}*t(min)*v(mL))$. Error was computed as standard deviation (SD). Data is from three biological replicates.

## Strains and plasmids construction

In-frame deletions were created using pNPTS138 or pK18*mobsacB* derivatives constructed as follows: pNPTS_Δ*neuB*: PCR was used to amplify two DNA fragments flanking the *neuB* ORF, by using primers neuB_ko1/neuB_ko2 and neuB_ko3/neuB_ko4. The PCR fragments were digested with *Eco*RI/*Bam*HI and *Bam*HI/*Hind*III, respectively, and ligated into pNPTS138 restricted with *Eco*RI and *Hind*III.

pSA37: PCR was used to amplify two DNA fragments flanking the *flmG* ORF, using primers flmG_ko1/flmG_ko2 and flmG_ko3/flmG_ko4. The PCR fragments were digested with *Hind*III/*Bam*HI and *Bam*HI/*Eco*RI, respectively, and ligated into pNPTS138 restricted with *Eco*RI and *Hind*III.

pSA252: PCR was used to amplify two DNA fragments flanking the *flmH* ORF, using primers flmH_ko1/flmH_ko2 and flmH_ko3/flmH_ko4. The PCR fragments were digested with *Eco*RI/*Xba*I and *Xba*I/*Hind*III, respectively, and ligated into pNPTS138 restricted with *Eco*RI and *Hind*III.

pSA265: PCR was used to amplify two DNA fragments flanking the *CCNA_01531* ORF, using primers 1531_ko1/1531_ko2 and 1531_ko3/1531_ko4. The PCR fragments were digested with *Eco*RI/*Bam*HI and *Bam*HI/*Hind*III, respectively, and ligated into pNPTS138 restricted with *Eco*RI and *Hind*III.

pSA617: PCR was used to amplify two DNA fragments flanking the *CCNA_01537* ORF, using primers 1537_ko1/1537_ko2 and 1537_ko3/1537_ko4. The PCR fragments were digested with *Nhe*I/*Bam*HI and *Bam*HI/*Hind*III, respectively, and ligated into pNPTS138 restricted with *Nhe*I and *Hind*III.

pSA253: PCR was used to amplify two DNA fragments flanking the *flmD* ORF, using primers flmD_ko1/flmD_ko2 and flmD_ko3/flmD_ko4. The PCR fragments were digested with *Eco*RI/*Bam*HI and *Bam*HI/*Hind*III, respectively, and ligated into pNPTS138 restricted with *Eco*RI and *Hind*III.

pSA35: PCR was used to amplify two DNA fragments flanking the *rkpQ* ORF, using primers rkpQ_ko1/rkpQ_ko2 and rkpQ_ko3/rkpQ_ko4. The PCR fragments were digested with *Eco*RI/*Bam*HI and *Bam*HI/*Hind*III, respectively, and ligated into pNPTS138 restricted with *Eco*RI and *Hind*III.

pSA326: PCR was used to amplify two DNA fragments flanking the *rkp3_013* ORF, using primers 013_ko1/013_ko2 and 013_ko3/013_ko4. The PCR fragments were digested with *Eco*RI/*Xba*I and

*Xba*I/*Hind*III, respectively, and ligated into pK18*mobsacB* (*Schäfer et al., 1994*) restricted with *Eco*RI and *Hind*III.

Bi-parental matings were used to transfer the resulting constructs into *C. crescentus* or *S. fredii* strains. Double recombination was selected by plating bacteria onto PYE (for *C. crescentus*) or TY (for *S. fredii*) plates containing 3% sucrose. Putative mutants were confirmed by PCR using primers external to the DNA fragments used for the in-frame deletion constructs.

Plasmids for constitutive expression (from P*van*, P*xyl*, or P*lac*) were constructed as follows: pSA53: *neuB* ORF was amplified by PCR with primers NeuB_N (with *Nde*I site overlapping the start codon) and NeuB_E (with *Eco*RI site flanking the stop codon) and cloned into pMT335, restricted with *Nde*I and *Eco*RI.

pSA59: *flmG* ORF was amplified by PCR with primers FlmG_N (with *Nde*I site overlapping the start codon) and FlmG_E (with *Eco*RI site flanking the stop codon) and cloned into pMT335, restricted with *Nde*I and *Eco*RI.

pSA645: the *flmG* sequence encoding the glycosyltransferase domain (residues 309–596) was amplified by PCR with primers FlmG_int_N and FlmG_E and cloned into pMT335, restricted with *Nde*I and *Eco*RI.

pSA126: *C. jejuni* 11168 *neuB1* ORF was amplified by PCR with primers NeuB_Cj1_N (with *Nde*I site overlapping the start codon) and NeuB_Cj1_E (with *Eco*RI site flanking the stop codon) and cloned into pMT335, restricted with *Nde*I and *Eco*RI.

pSA47: *C. jejuni* 11168 *neuB2* ORF was amplified by PCR with primers NeuB_Cj2_N (with *Nde*I site overlapping the start codon) and NeuB_Cj2_E (with *Eco*RI site flanking the stop codon) and cloned into pMT335, restricted with *Nde*I and *Eco*RI.

pSA48: *C. jejuni* 11168 *neuB3* ORF was amplified by PCR with primers NeuB_Cj3_N (with *Nde*I site overlapping the start codon) and NeuB_Cj3_E (with *Eco*RI site flanking the stop codon) and cloned into pMT335, restricted with *Nde*I and *Eco*RI.

pSA60: *fljK* ORF was amplified by PCR with primers FljK_N (with *Nde*I site overlapping the start codon) and FljK_X (with *Xba*I site flanking the stop codon) and cloned into pMT335, restricted with *Nde*I and *Xba*I.

pSA104: *fljK* ORF was subcloned from pSA60 into pMT463, using *Nde*I and *Xba*I.

pSA42: *S. fredii* NGR234 *rkpQ* ORF was amplified by PCR with primers RkpQ_N (with *Nde*I site overlapping the start codon) and RkpQ_E (with *Eco*RI site flanking the stop codon) and cloned into pMT335, restricted with *Nde*I and *Eco*RI.

pSA263: *S. fredii* NGR234 *rkpO* ORF was amplified by PCR with primers RkpO_N (with *Nde*I site overlapping the start codon) and RkpO_M (with *Mun*I site flanking the stop codon) and cloned into pMT335, restricted with *Nde*I and *Eco*RI.

pSA569: *S. fredii* NGR234 *rkpL* ORF was amplified by PCR with primers RkpL_N (with *Nde*I site overlapping the start codon) and RkpL_E (with *Eco*RI site flanking the stop codon) and cloned into pMT335, restricted with *Nde*I and *Eco*RI.

pSA568: *S. fredii* NGR234 *rkpM* ORF was amplified by PCR with primers RkpM_N (with *Nde*I site overlapping the start codon) and RkpM_E (with *Eco*RI site flanking the stop codon) and cloned into pMT335, restricted with *Nde*I and *Eco*RI.

pSA570: *S. fredii* NGR234 *rkpLM* operon was amplified by PCR with primers RkpL_N (with *Nde*I site overlapping the *rkpL* start codon) and RkpM_E (with *Eco*RI site flanking the *rkpM* stop codon) and cloned into pMT335, restricted with *Nde*I and *Eco*RI.

To create pSA454, a synthetic fragment encoding *C. crescentus fljK* (sequence optimized for *E. coli*, *Supplementary file 3* Table S3) was ligated into pSRK-Gm (*Khan et al., 2008*), using *Nde*I and *Xba*I.

To create pSA496, the *flmG* ORF was subcloned from pSA59 into pSRK-Gm, using *Nde*I and *Xba*I.

To create the vector for co-expression of *fljK* and *flmG* in *S. fredii*, the synthetic fragment encoding *C. crescentus fljK* ORF was first ligated into pMT335, using *Nde*I and *Eco*RI, creating pSA107. Then *flmG* ORF was amplified by PCR with primers FlmG_rbs_E (with ribosome binding site and *Eco*RI site flanking the *flmG* start codon) and FlmG_X (with *Xba*I site flanking the *flmG* stop codon) and cloned into pSA107, restricted with *Eco*RI and *Xba*I, creating pSA235. Finally, the *fljK* and *flmG* ORFs were subcloned from pSA235 into pSRK-Gm, using *Nde*I and *Xba*I, creating pSA236.

pSA571: *flmH* ORF was amplified by PCR with primers FlmH_N (with *Nde*I site overlapping the start codon) and FlmH_E (with *Eco*RI site flanking the stop codon) and cloned into pMT335, restricted with *Nde*I and *Eco*RI. The *flmH* ORF was then subcloned from pMT335 into pSRK-Km (*Khan et al., 2008*), using *Nde*I and *Xba*I.

pSA572: *CCNA_01531* ORF was amplified by PCR with primers 1531_N (with *Nde*I site overlapping the start codon) and 1531_E (with *Eco*RI site flanking the stop codon) and cloned into pMT335, restricted with *Nde*I and *Eco*RI. The *CCNA_01531* ORF was then subcloned from pMT335 into pSRK-Km, using *Nde*I and *Xba*I.

pSA624: *CCNA_01537* ORF was amplified by PCR with primers 1537_N (with *Nde*I site overlapping the start codon) and 1537_X (with *Xba*I site flanking the stop codon) and cloned into pSRK-Km, restricted with *Nde*I and *Xba*I.

pSA573: *S fredii* NGR234 *rkp3_013* ORF was amplified by PCR with primers rkp3_013_N (with *Nde*I site overlapping the start codon) and rkp3_013_M (with *Mun*I site flanking the stop codon) and cloned into pMT335, restricted with *Nde*I and *Eco*RI. The *rkp3_013* ORF was then subcloned from pMT335 into pSRK-Km, using *Nde*I and *Xba*I.

For *neuB* site-directed mutagenesis:

pSA58: *neuB* ORF was subcloned from pSA53 into pOK12, using *Nde*I and *Eco*RI. pSA58 was used as a template for site-directed mutagenesis of *neuB*.

pSA90: pOK12 carrying *neuB* E30A allele.

pSA91: pOK12 carrying *neuB* H245A allele.

pSA92: pOK12 carrying *neuB* R322A allele.

pSA93: the *neuB* E30A allele was subcloned from pSA90 into pMT335, using *Nde*I and *Eco*RI.

pSA94: the *neuB* H245A allele was subcloned from pSA91 into pMT335, using *Nde*I and *Eco*RI.

pSA95: the *neuB* R322A allele was subcloned from pSA92 into pMT335, using *Nde*I and *Eco*RI.

pIDT-*flm*: to express the complete *C. crescentus* pseudaminic acid biosynthesis pathway in *E. coli*, a synthetic sequence (codon-optimized for *E. coli*; *Supplementary file 3* Table S3) encoding for *flmA* (*CCNA_00233*), *flmB* (*CCNA_00234*), *flmH* (*CCNA_01523*), *flmD* (*CCNA_02947*), *neuB* (*CCNA_02961*) and *flmC* (*CCNA_02946*) was expressed from pUCIDT under control of the T5 promoter.

Plasmid pSA44 is a derivative of pET28a expressing $His_6$-NeuB under control of the T7 promoter. To construct pSA44, *neuB* ORF was amplified by PCR with primers NeuB_N (with *Nde*I site overlapping the start codon) and NeuB_E (with *Eco*RI site flanking the stop codon) and cloned into pET28a, restricted with *Nde*I and *Eco*RI.

To express FljK in *E. coli* without $His_6$ tag, *fljK* ORF was subcloned from pMT335 into pET47b, using *Nde*I and *Sac*I, creating pSA106.

Plasmid pSA363 is a derivative of pCWR547 (*Radhakrishnan et al., 2010*) expressing $His_6$-SUMO-FlmG$_{(301-500)}$ under control of the T7 promoter. To construct pSA363, a fragment encoding residues 301–500 of FlmG was amplified by PCR with primers FlmG_547_N and FlmG_547_S, digested with *Nde*I and *Sac*I and cloned into pCWR547 restricted with the same enzymes.

Plasmids used for the BACTH experiments were constructed as follows: pNK92: the sequence encompassing the last 288 amino acids of FlmG containing the glycosyltransferase domain (GT) was amplified with primers NK-77 and NK-75 and cloned into pKNT25 restricted with *Xba*I and *Kpn*I.

pNK93: the sequence encompassing the last 288 amino acids of FlmG containing the glycosyltransferase domain (GT) was amplified with primers NK-77 and NK-79 and cloned into pKT25 restricted with *Xba*I and *Kpn*I.

pNK95: *flmG* ORF was amplified with primers NK-74 and NK-75 and cloned into pKNT25 restricted with *Xba*I and *Kpn*I.

pNK96: *flmG* ORF was amplified with primers NK-74 and NK-79 and cloned into pKT25 restricted with *Xba*I and *Kpn*I. pNK98: the sequence encompassing the first 313 amino acids of FlmG containing the TPR domain was amplified with primers NK-74 and NK-76 and cloned into pKNT25 restricted with *Xba*I and *Kpn*I.

pNK99: the sequence encompassing the first 313 amino acids of FlmG containing the TPR domain was amplified with primers NK-74 and NK-78 and cloned into pKT25 restricted with *Xba*I and *Kpn*I.

pNK16: *fljJ* ORF was amplified with primers NK-01 and NK-02 and cloned into pUT18C restricted with *Xba*I and *Kpn*I.

pNK144: *fljK* ORF was amplified with primers NK-10 and NK-11 and cloned into pUT18C restricted with *Xba*I and *Kpn*I.

pNK330: *fljM* ORF was amplified with primers NK-62 and NK-64 and cloned into pUT18C restricted with *Xba*I and *Kpn*I.

To create *lacZ* transcriptional fusions, promoter regions were cloned into pRKlac290 (*Gober and Shapiro, 1992*) as follows: pRKlac290_P*neuB*: a 569 bp DNA fragment was amplified by PCR with primers PneuB_E/PneuB_X, digested with *Eco*RI and *Xba*I, and ligated into pRKlac290, cut with the same enzymes.

pRKlac290_P*flmG*: a 579 bp DNA fragment was amplified by PCR with primers PflmG_E/PflmG_X, digested with *Eco*RI and *Xba*I, and ligated into pRKlac290, cut with the same enzymes.

pRKlac290_P*flmA*: a 570 bp DNA fragment was amplified by PCR with primers PflmA_E/PflmA_X, digested with *Eco*RI and *Xba*I, and ligated into pRKlac290, cut with the same enzymes.

# Acknowledgements

We thank Laurence Degeorges for excellent technical assistance and acknowledge the Swiss National Science Foundation grant 31003A_182576 for the funding support.

# Additional information

## Competing interests

Patrick H Viollier: The authors declare a pending patent application PAT7460EP00 on FlmG-dependent soluble protein glycosylation systems in bacteria. The other authors declare that no competing interests exist.

## Funding

| Funder | Grant reference number | Author |
|---|---|---|
| Swiss National Science Foundation | 31003A_182576 | Patrick H Viollier |

The funders had no role in study design, data collection and interpretation, or the decision to submit the work for publication.

## Author contributions

Silvia Ardissone, Nicolas Kint, Conceptualization, Resources, Investigation, Methodology, Writing - original draft, Writing - review and editing; Patrick H Viollier, Conceptualization, Funding acquisition, Writing - original draft, Writing - review and editing

## Author ORCIDs

Silvia Ardissone https://orcid.org/0000-0003-4346-8124
Nicolas Kint https://orcid.org/0000-0001-5905-2639
Patrick H Viollier https://orcid.org/0000-0002-5249-9910

## Decision letter and Author response

Decision letter https://doi.org/10.7554/eLife.60488.sa1
Author response https://doi.org/10.7554/eLife.60488.sa2

# Additional files

## Supplementary files

- Supplementary file 1. Table S1. LacZ activity of BACTH assay.

- Supplementary file 2. Table S2. Strains and plasmids used in this study.

- Supplementary file 3. Table S3. Oligonucleotides and synthetic genes used in this study.

• Transparent reporting form

### Data availability

All data generated or analysed during this study are included in the manuscript and supporting files.

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

# Appendix 1

**Appendix 1—key resources table**

| Reagent type (species) or resource | Designation | Source or reference | Identifiers | Additional information |
|---|---|---|---|---|
| Strain, Strain Background (*Caulobacter crescentus* NA1000) | *Caulobacter crescentus* NA1000 | *Evinger and Agabian, 1977*; PMID:334726 | | See *Supplementary file 2* Table S2 |
| Antibody | CtrA Rabbit polyclonal | *Domian et al., 1997*; PMID:9267022 | Home-made antibodies raised against full-length protein of *C. crescentus* | Immunoblot: 1/10000 dilution *Figure 6*, *Figure 6— figure supplement 1* |
| Antibody | FlgE Rabbit polyclonal | *Hahnenberger and Shapiro, 1987*; PMID:3039149 | Home-made antibodies raised against FlgE protein of *C. crescentus* | Immunoblot: 1/50000 dilution *Figure 1* |
| Antibody | Flagellins Rabbit polyclonal | *Hahnenberger and Shapiro, 1987*; PMID:3039149 | Home-made antibodies raised against purified flagellins from *C. crescentus* | Immunoblot: 1/20000 dilution *Figure 1*, *Figure 2*, *Figure 3*, *Figure 4*, *Figure 5*, *Figure 6— figure supplement 1* |
| Antibody | FljK Rabbit polyclonal | *Ardissone et al., 2020* | Home-made antibodies raised against full-length protein of *C. crescentus* | Immunoblot: 1/10000 dilution *Figure 1*, *Figure 4*, *Figure 5*, *Figure 5—figure supplement 1* |
| Antibody | NeuB Rabbit polyclonal | This paper | Home-made antibodies raised against full-length protein of *C. crescentus* | Immunoblot: 1/10000 dilution *Figure 2*, *Figure 3*, *Figure 6—figure supplement 1* |
| Antibody | FlmG Rabbit polyclonal | This paper | Home-made antibodies raised against FlmG (amino acids 301–500) of *C. crescentus* | Immunoblot: 1/10000 dilution *Figure 6* |
| Recombinant DNA reagent | Plasmids | This paper | | See *Supplementary file 2* Table S2 |
| Sequence-based reagent | PCR primers | This paper | | See *Supplementary file 3* Table S3 |
| Sequence-based reagent | FljK synthetic sequence | This paper | FljK coding sequence, codon-optimized for *E. coli* | See *Supplementary file 3* Table S3 |
| Sequence-based reagent | Pseudaminic acid biosynthesis operon synthetic sequence | This paper | FlmA, FlmB, FlmH, FlmD, NeuB and FlmC coding sequences, codon-optimized for *E. coli* | See *Supplementary file 3* Table S3 |

