## [Decision Letter]

Thank you for submitting your article "Specificity in glycosylation of multiple flagellins by the modular and cell cycle regulated glycosyltransferase FlmG" for consideration by *eLife*. Your article has been reviewed by two peer reviewers, one of whom is a member of our Board of Reviewing Editors, and the evaluation has been overseen by Gisela Storz as the Senior Editor. The reviewers have opted to remain anonymous.

The reviewers have discussed the reviews with one another and the Reviewing Editor has drafted this decision to help you prepare a revised submission.

Summary:

Both reviewers agree that this manuscript is suitable for publication in e*Life* after minor revision. The authors unravel the pathway which leads to O-glycoslyation of the six flagellins in *Caulobacter crescentus* and show that O-glycosylation of the flagellins is essential for flagella assembly.

Essential revisions:

A large open question is why flaggelin glycosylation is required for filament assembly. Moreover, the authors should at least discuss in more detail the sometimes contradicting results (e.g. FljK is no longer secreted in the neuB point mutants compared to a neuB deletion strain).

Reviewer #1:

In this manuscript the authors delineate the pathway which O-glycosylates the flagellins in *C. crescentus*. Using genetics, they show that NeuB is involved in the biosynthesis of pseudaminic acid, which is transferred by FlmG onto the flagellins. FlmG has to domains: the N-terminal domains binds specifically to the different flagellins whereas the C-terminal domain transfers the sugar moiety to the flagellins. Furthermore, all enzymes essential for the biosynthesis of pseudaminic acid are identified using deletion mutant screens.

The manuscript is very well written. However, I am not sure why the authors emphasize so much on the possible use of FlmG for biotechnological application in O-glycosylation. This is discussed in the last sentence of the Abstract and not necessary there. It would be great to finish the Abstract with the importance for the cell cycle regulation in *C. crescentus* as done in the cover letter. And the same is true for the Discussion of the manuscript: the first two pages discuss the pro- and cons of developing a biotech model for O-glycosylation using FlmG. I would see this as a last paragraph and outlook at the end of the Discussion in a much-shortened form.

The authors ask the question how FlmG and FlaF compete for flagellin binding. Maybe I overlooked it, but did the authors test whether FlaF binds both non-glycosylated and glycosylated flagellins ? If it would only bind glycosylated flagellins that this would ensure the timing of binding and therefore export of only glycosylated flagellins.

Reviewer #2:

The flagella of many bacteria are post-translationally modified, facilitating e.g. assembly of the flagellar filament, flagella-mediated surface adhesion and immune system evasion.

In the present manuscript, Ardissone and colleagues investigated the O-glycosylation pathway that modifies six flagellin paralogs in *Caulobacter crescentus*. They found that the pseudaminic acid pathway is required for motility, flagellation and modification of flagellin. Analyzing a Tn library of motility mutants revealed an hitherto uncharacterized O-linking glycosyltransferase (FlmG) required for flagellin modification in the presence of pseudaminic acid. They further found that protein levels of FlmG peak at the end of the cell cycle, while the FlaF secretion chaperone accumulates in dividing cells. This suggests a mechanism where flagellin modification by FlmG can occur before the FlaF-dependent secretion, thus avoiding competition for flagellin binding.

Post-translational protein modifications are essential for many cellular functions, but the underlying substrate selectivity remains poorly understood. The present manuscript makes an important contribution towards our understanding of flagellin glycosylation in *Caulobacter crescentus*. The work is elegantly carried out and the manuscript is well written. The experiments mostly support the author's conclusions, but some concerns remain. I have the following comments and suggestions to improve the author's conclusions and the clarity of the manuscript.

1) Glycosylation of flagellin is required for filament assembly in Caulobacter and the authors speculate in their Discussion that another assembly factor might be glycosylated and required for filament assembly. However, Figure 1E shows that unmodified flagellin is only poorly secreted, which suggests that glycosylation of flagellin is needed for efficient secretion and/or recognition by the export apparatus?

2) Figure 1E: The Western blot is overexposed, but it appears that two species of modified FljK exist? However, only one of the two species is secreted? How many serine/threonine residues are modified in Caulobacter FljK?

3) Figure 2C: Why is FljK not secreted anymore in the neuB point mutants compared to a neuB deletion strain (e.g. compare Figure 1E)? Does this suggest a direct interaction of NeuB with flagellin and the catalytically-inactive NeuB point mutants might thereby prevent FljK secretion?

---

## [Author Response]

Reviewer #1:In this manuscript the authors delineate the pathway which O-glycosylates the flagellins in C. crescentus. Using genetics, they show that NeuB is involved in the biosynthesis of pseudaminic acid, which is transferred by FlmG onto the flagellins. FlmG has to domains: the N-terminal domains binds specifically to the different flagellins whereas the C-terminal domain transfers the sugar moiety to the flagellins. Furthermore, all enzymes essential for the biosynthesis of pseudaminic acid are identified using deletion mutant screens.The manuscript is very well written. However, I am not sure why the authors emphasize so much on the possible use of FlmG for biotechnological application in O-glycosylation. This is discussed in the last sentence of the Abstract and not necessary there. It would be great to finish the Abstract with the importance for the cell cycle regulation in C. crescentus as done in the cover letter. And the same is true for the Discussion of the manuscript: the first two pages discuss the pro- and cons of developing a biotech model for O-glycosylation using FlmG. I would see this as a last paragraph and outlook at the end of the Discussion in a much-shortened form.

We agree and have modified the Abstract and Discussion accordingly.

The authors ask the question how FlmG and FlaF compete for flagellin binding. Maybe I overlooked it, but did the authors test whether FlaF binds both non-glycosylated and glycosylated flagellins ? If it would only bind glycosylated flagellins that this would ensure the timing of binding and therefore export of only glycosylated flagellins.

The BACTH experiments were conducted in *E. coli* K12 (which does not produce pseudaminic acid and thus only unmodified [non-glycosylated] flagellin even in the presence of FlmG, see Figure 1F) and showed (in the companion manuscript) that the secretion chaperone FlaF can bind flagellins in the absence of glycosylation. However, we cannot exclude that glycosylation would enhance the binding of FlaF to flagellins, but our *E. coli*-based BACTH system does not permit expression of glycosylated versus non-glycosylated variants compatible with the *E. coli*-based BACTH assay.

Reviewer #2:The flagella of many bacteria are post-translationally modified, facilitating e.g. assembly of the flagellar filament, flagella-mediated surface adhesion and immune system evasion.In the present manuscript, Ardissone and colleagues investigated the O-glycosylation pathway that modifies six flagellin paralogs in Caulobacter crescentus. They found that the pseudaminic acid pathway is required for motility, flagellation and modification of flagellin. Analyzing a Tn library of motility mutants revealed an hitherto uncharacterized O-linking glycosyltransferase (FlmG) required for flagellin modification in the presence of pseudaminic acid. They further found that protein levels of FlmG peak at the end of the cell cycle, while the FlaF secretion chaperone accumulates in dividing cells. This suggests a mechanism where flagellin modification by FlmG can occur before the FlaF-dependent secretion, thus avoiding competition for flagellin binding.Post-translational protein modifications are essential for many cellular functions, but the underlying substrate selectivity remains poorly understood. The present manuscript makes an important contribution towards our understanding of flagellin glycosylation in Caulobacter crescentus. The work is elegantly carried out and the manuscript is well written. The experiments mostly support the author's conclusions, but some concerns remain. I have the following comments and suggestions to improve the author's conclusions and the clarity of the manuscript.1) Glycosylation of flagellin is required for filament assembly in Caulobacter and the authors speculate in their Discussion that another assembly factor might be glycosylated and required for filament assembly. However, Figure 1E shows that unmodified flagellin is only poorly secreted, which suggests that glycosylation of flagellin is needed for efficient secretion and/or recognition by the export apparatus?

We revised the Discussion to better address these issues. In a nutshell, yes flagellin secretion is likely reduced, however we cannot rule out that other events in assembly are also defective and we prefer not to make a definitive statement about secretion.

2) Figure 1E: The Western blot is overexposed, but it appears that two species of modified FljK exist? However, only one of the two species is secreted? How many serine/threonine residues are modified in Caulobacter FljK?

Four threonines have recently been reported to be modified in FljK, see Montemayor et al., 2020.

3) Figure 2C: Why is FljK not secreted anymore in the neuB point mutants compared to a neuB deletion strain (e.g. compare Figure 1E)? Does this suggest a direct interaction of NeuB with flagellin and the catalytically-inactive NeuB point mutants might thereby prevent FljK secretion?

The neuB deletion mutant (vector control) is on the same blot and shows the same effect as the two (non-functional) missense mutants